# Risk Assessment of Rising Temperatures Using Landsat 4–9 LST Time Series and Meta® Population Dataset: An Application in Aosta Valley, NW Italy

Tommaso Orusa [1,2,3,*], Annalisa Viani [4], Boineelo Moyo [5,6], Duke Cammareri [2,3] and Enrico Borgogno-Mondino [1]

1 Department of Agricultural, Forest and Food Sciences (DISAFA), GEO4Agri DISAFA Lab, Università degli Studi di Torino, Largo Paolo Braccini 2, 10095 Grugliasco, Italy; enrico.borgogno@unito.it
2 Earth Observation Valle d'Aosta—eoVdA, Località L'Île-Blonde, 5, 11020 Brissogne, Italy; dcammareri@invallee.it
3 IN.VA spa, Località L'Île-Blonde, 5, 11020 Brissogne, Italy
4 Istituto Zooprofilattico Sperimentale Piemonte, Liguria e Valle d'Aosta (IZS PLV) S.C Valle d'Aosta—CeRMAS (National Reference Center for Wildlife Diseases), Località Amerique, 7/G, 11020 Quart, Italy; annalisa.viani@izsto.it
5 Department of Geoinformatics, Stuttgart University of Applied Sciences, Schellingstraße 24, 70174 Stuttgart, Germany
6 HRSL, Core Science, Meta Research, 1 Hacker Way, Menlo Park, CA 94025, USA
* Correspondence: tommaso.orusa@unito.it

**Abstract:** Earth observation data have assumed a key role in environmental monitoring, as well as in risk assessment. Rising temperatures and consequently heat waves due to ongoing climate change represent an important risk considering the population, as well as animals, exposed. This study was focused on the Aosta Valley Region in NW Italy. To assess population exposure to these patterns, the following datasets have been considered: (1) HDX Meta population dataset refined and updated in order to map population distribution and its features; (2) Landsat collection (missions 4 to 9) from 1984 to 2022 obtained and calibrated in Google Earth Engine to model LST trends. A pixel-based analysis was performed considering Aosta Valley settlements and relative population distribution according to the Meta population dataset. From Landsat data, LST trends were modelled. The LST gains computed were used to produce risk exposure maps considering the population distribution and structure (such as ages, gender, etc.). To check the consistency and quality of the HDX population dataset, MAE was computed considering the ISTAT population dataset at the municipality level. Exposure-risk maps were finally realized adopting two different approaches. The first one considers only LST gain maximum by performing an ISODATA unsupervised classification clustering in which the separability of each class obtained and was checked by computing the Jeffries–Matusita (J-M) distances. The second one was to map the rising temperature exposure by developing and performing a risk geo-analysis. In this last case the input parameters considered were defined after performing a multivariate regression in which LST maximum was correlated and tested considering (a) Fractional Vegetation Cover (FVC), (b) Quote, (c) Slope, (d) Aspect, (e) Potential Incoming Solar Radiation (mean sunlight duration in the meteorological summer season), and (f) LST gain mean. Results show a steeper increase in LST maximum trend, especially in the bottom valley municipalities, and especially in new built-up areas, where more than 60% of the Aosta Valley population and domestic animals live and where a high exposure has been detected and mapped with both approaches performed. Maps produced may help the local planners and the civil protection services to face global warming from a One Health perspective.

**Keywords:** Google Earth Engine; USGS NASA Landsat 4–9 missions; LST timeseries analysis; risk population assessment; HDX meta population; trends modeling; Aosta Valley; Italy; Alps; climate change





## 1. Introduction

Temperature and summer heatwave monitoring due to ongoing climate change has assumed a crucial role in the last years worldwide [1–3]. Although studies on extreme events are increasing, and in particular on heatwaves and urban heat islands [4–8], few focus on time series derived from free Earth observation images [4,9–15]. Furthermore, there is still a lack regarding scientific and technical studies that focus on land surface temperature (hereinafter called LST) climatic trends through an analysis of the exposed population and related risks [16–18].

Nowadays, many studies focus on LST and epidemiological relationships but do not concern themselves with spatial population exposure [19–21] or animals, including wildlife [22].

### 1.1. Earth Observation (EO) Data Role in the Climate Change Framework

The evaluation of exposure to ambient temperatures in epidemiological studies has generally been based on records from meteorological stations which may not adequately represent local temperature variability [23].

To go beyond this limiting factor, Earth observation images represent a possible solution to carefully map environmental conditions at the local scale [20,24]. The health sector and civil protection services in recent years at the European, Italian, and local levels are particularly interested in having cartographic products and GIS to assess the risks and effects of extreme temperatures on the population by identifying the most vulnerable areas [19]. The identification of these areas would make it possible to direct territorial planning towards greening policies or measures aimed at mitigating warming and at the same time implementing forms of adaptation (for example, creation of emergency response hubs in the case of an area with a vulnerable population such as the elderly). Although free thermal data are increasing by offering medium-high spatial resolution (such as Landsat missions [25,26] with a resampled GSD of 30 m or ECOSTRESS with 60 m GSD [27–32], their use for the development of various services and applications is still limited [33–36] and therefore, offer numerous exploitation possibilities when combined with new databases made available by various governmental or research bodies.

Thermal data are widely applied nowadays to map LST and urban heat island phenomena [8,37–41]. However, their use is often confined to analysis at given moments and not in timeseries due to the need to calibrate them [34,42]. Platforms such as Google Earth Engine [43] in the case of Landsat data allow, thanks to the algorithm developed by Ermida [36], to quickly calibrate the thermal data allowing analysis on historical series.

### 1.2. Population Datasets

In recent years, datasets on the spatial and temporal distribution of the global population have been developed [44–46]. Nevertheless, they still have a coarse resolution. One of the most detailed is provided by the World Bank with the World Population dataset with a spatial resolution of 1 km and another of 100 m. This last is spatially coeval with the native geometric resolution for the thermal bands of the Landsat missions [46]. This dataset contains a top-down constrained breakdown of estimated population by age and gender groups from 2000 to the present year [45]. Top-down constrained age/sex structure estimate datasets for individual countries for 2020 at 100 m spatial resolution with country totals adjusted to match the corresponding official United Nations population estimates have been prepared by the Population Division of the Department of Economic and Social Affairs of the United Nations Secretariat (2019 Revision of World Population Prospects). It is worth noting that WorldPop gridded datasets on population age structures, poverty, urban growth, and population dynamics are freely available. Despite the huge amount of data, this dataset still has limited application in rural contexts and outside wide urban areas due to its geometric resolution that has limited the application at regional and local levels [47–49].

Accurate information on global population distribution is crucial to many disciplines. A population and housing census is the traditional tool for deriving small-area detailed statistics on population and its spatial distribution [50,51]. However, censuses are time-consuming, and the spatial resolution is naturally set by the census enumeration areas (EA), which lack fine-grained information about the aggregation of population. The sizes of the EAs vary by many orders of magnitude from country to country, ranging from hundreds of square meters in urban areas to tens of thousands of square kilometers in low-population areas, resulting in an average spatial resolution [50] of a census unit of 33 km at a global scale. Recently, multiple higher-resolution maps of human-made built-up areas have emerged [52,53], most notably the Global Human Settlement Layer (GHSL) [54], the Global Urban Footprint (GUF) [51,55], the WorldPop project [44,56], Landscan [57,58], and Missing Maps project [59,60]. However, none provide a scalable solution with high accuracy in rural areas. Over the last decade high-resolution (sub-meter) satellite imagery has become widely available, enabling the global collection of recent and cloud-free Earth imagery. Additionally, in the last years, the surge in research on computer vision and in particular convolutional neural networks (CNNs) have enabled bulk processing of imagery in a rapid manner [50]. The combination of these methods enables the global analysis of high-resolution imagery as a promising method for detecting individual buildings; combining building estimates with available census data to produce updated and higher-resolution population maps; and offering alternative, state-of-the-art population estimates in the absence of census data. Various approaches using machine learning have been demonstrated on small areas [50], yet a method which allows global mapping has remained elusive.

Nowadays, in fact, high-resolution datasets of population density which accurately map sparsely-distributed human populations do not exist at a global scale [49,50,61]. Typically, population data are obtained using censuses and statistical modeling. More recently, methods using remotely-sensed data have emerged, capable of effectively identifying urbanized areas. Obtaining high accuracy in the estimation of population distribution in rural areas remains a very challenging task due to the simultaneous requirements of sufficient sensitivity and resolution to detect very sparse populations through remote sensing as well as reliable performance at a global scale. Meta has recently developed a computer vision method based on machine learning to create population maps from satellite imagery and phone GNSS tracking at a global scale, with a spatial sensitivity corresponding to individual buildings and suitable for global deployment. By combining these settlement data with census data, Meta has created the HDX Meta population dataset, including raster maps with ~30 m spatial resolution for 18 countries in the world [50]. HDX is a platform which lets users, for research and management purposes, access socio-economic data mostly collected by Meta through Data for Good (https://dataforgood.facebook.com/dfg, last accessed on 18 April 2023). Data for Good at Meta's program includes tools built from de-identified Meta data, as well as tools that the company develops using satellite imagery and other publicly available sources.

### 1.3. Remote Sensing in Climate Change Risk Assessment

There is a growing need for the assessment and reduction of climate change risk. The effects of global warming are already bringing harm to human communities and the natural world. Further temperature rises will have a devastating impact and more action on greenhouse gas emissions is urgently required. Multiple factors contribute to climate change, and multiple actions are needed to address it [8,37–41]. Especially concerning is population exposure to climate change. In fact, nowadays, EO data and more generally remote sensing may help in mapping and developing services with particular regard to climate change risk assessment involving communities at different levels. Space-borne images for civil applications have been routinely acquired since the 1980s (Landsat and SPOT), while more recently, the European Union's Copernicus program has been acquiring images. EO data can provide remotely sensed information regarding floods, forest fires, and droughts. In general, remote sensing data from space, but also from airborne or drone

platforms, can be profitably used to manage many risks, from geo-hydrological to volcanic, and from seismic to anthropogenic. Less exploited is the application and coupling of remote sensing data with GIS health data, with particular regard to the climate change framework. Remote sensing can play a key role in managing risks, leading to a new level of understanding of the complex Earth systems and planning. In recent decades, satellite-based observations and the derived geospatial products have been successfully demonstrated to be highly valuable tools in each different phase of the risk and exposure management (forecasting, planning, emergency, and post-emergency) [34,42]. For example, synthetic aperture radar (SAR) images can facilitate risk management since they are also acquired through dense cloud cover and in both night and day conditions. This ability can help during the emergency phase. Stacks of SAR data can be used to detect subtle ground deformation induced by slow movement phenomena (e.g., slow landslides, subsidence) that can dangerously evolve, involving elements of risk [62]. On the other hand, optical images are fundamental products for monitoring land cover changes induced by several hazards (e.g., fast landslides, volcanic eruptions) or thermal data to assess, for example, urban heath islands (UHIs) and their intensity or the water stress on forests and crops. These data are routinely used to map and evaluate the elements at risk scattered over wide areas. The application of a combined use of population data at higher resolution with thermal EO data in order to evaluate the exposure to rising temperature in light of climate change has been poorly explored in the scientific community. This is due to the fact that the population datasets at higher resolution are relatively new, as is the application of EO data in the domain. For the climate change adaptation regarding the civil component, the first steps are being taken.

Moreover, the One Health approach involving thermal remote-sensed time-series analysis to assess the temperature trend gain represents a novelty compared with the well-known and over-studied UHI phenomenon which is focused only on a given time and does not permit the development of strong models. The LST trends analysis modelling and its application to coupling population data represent a novelty especially in the assessment of rising temperature exposure [8].

### 1.4. Coupling Population and EO Data in Climate Change Adaptation and Risk Assessment

The approach developed to map population (thanks to Meta Geo for Good) combined with free thermal EO data to model rising temperature represents a first attempt of this kind. In order to map the population exposure and risk for the first time, the highest available population dataset has been used, with a native geometrical resolution (GSD—Ground Sample Distance 30 m), which is the same as Landsat's. This may enforce the applicability and coupling of these kinds of data in the planning and management of climate change risks and adaptation, suggesting new solutions [18,63]. Furthermore, mapping the exposure of population involved according to different levels of temperature (LST) gain permits greening actions and policies to be addressed, favors the identification of new medical or health centers, permits the areas that will be more subject to emergency calls to be known in advance, allows areas or zones most at risk to be redeveloped with a view to mitigation and above all adaptation, makes forecasts on access to hospitals in case of heat waves and the impact of costs on the health sector having mapped data, and evaluates the effectiveness of requalification policies and actions and its effects on heat flows and on the risk associated with the exposed population. Then, the development of new applications and services in a technological perspective can help the transfer also to other sectors.

### 1.5. Aims

Finally, the main aim of this work was to perform a risk population assessment on rising temperatures and heat waves by Landsat LST timeseries in Aosta Valley, NW Italy by realizing a scalable application to all 18 countries that already have an HDX Meta dataset. The analysis was performed at a pixel level, grouping the final population exposure at a municipality level. It is worth noting that the map generated will be available at a pixel level.

Moreover, the quality of the population dataset was checked, and a risk map performed considering the population distribution and the LST gains modelized. In particular, LST maximum and mean trends were computed considering their significance, and possible correlations were tested considering the following parameters: (a) fractional vegetation cover (FVC), (b) quote, (c) slope, (d) aspect, and (e) potential incoming solar radiation (mean sunlight duration in the meteorological summer season) in order to assess which parameters include in the risk model.

The final outputs have permitted the mapping and assessment of the LST gain in the last 39 years (1984–2022) and relative population exposure to LST trends per age bands and gender groups, providing hopefully useful information to civil protection services and the health sector, permitting them to detect areas in which calls to health emergencies would be more likely during heatwaves and allowing urban planners to promote greening actions in a mitigation and adaptation perspective to climate change according to a One Health approach.

## 2. Materials and Methods

### 2.1. Aosta Valley Study Area

The study was carried out considering the Aosta Valley Autonomous Region in the northwest of Italy (please see Figure 1 below). To perform zonal statistics on the study territory, ESRI shapefile municipality boundaries were downloaded from the SCT Geoportale della Valle d'Aosta (https://geoportale.regione.vda.it/, last accessed on 22 March 2023) and adopted for the computation.

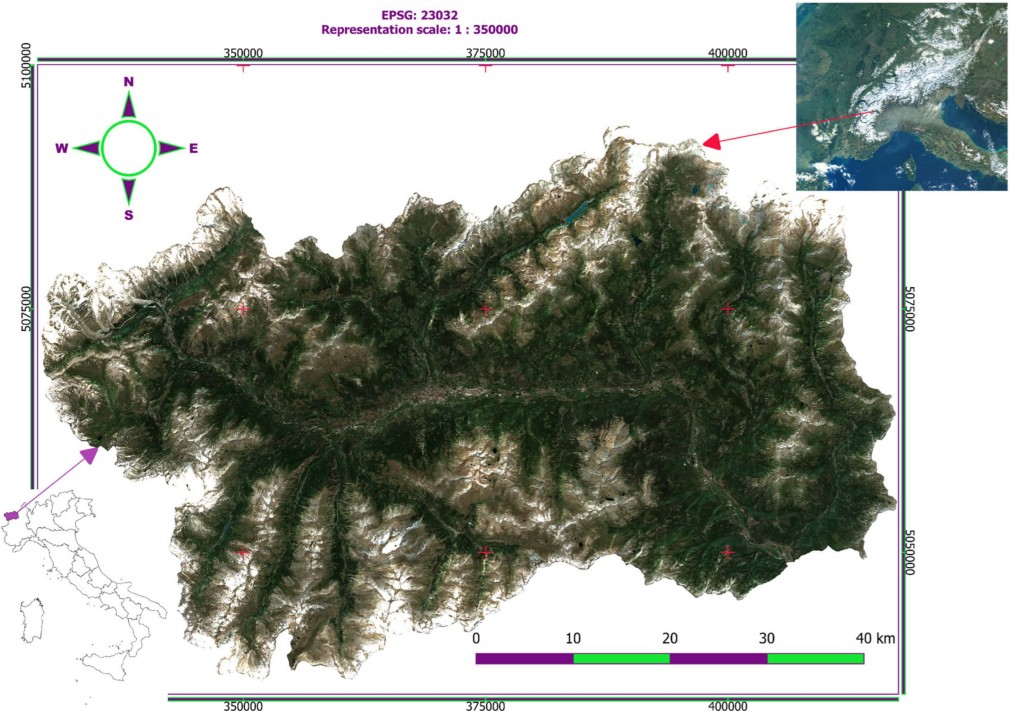

**Figure 1.** Study Area corresponding to the boundaries of the Aosta Valley francophone Autonomous Region (NW Italy).

### 2.2. Landsat Timeseries Datasets and LST Processing

LSTs have been computed from all Landsat missions that have been processed in Google Earth Engine (GEE). The United States Geological Survey (USGS) provides TOA brightness temperature images (hereinafter called Tb) as obtained from the thermal sensors of Landsat satellite missions. USGS Tb images collection 1 (from Landsat 4–5–7–8–9 missions, sensors TM, ETM+, TIRS) can be accessed through GEE. USGS also provides the corresponding at-the-surface reflectance-calibrated bands that can be, similarly, accessed

through GEE. Additionally, the quality-assessment band (BQA) is available too, making it possible to retrieve cloud coverage and shadowing information. All the bands are supplied with a ground sampling distance (GSD) of 30 m. Coarser thermal bands that natively have 100 m GSD have been oversampled by NASA using a bilinear approach at 30 m. The following GEE thermal collections were adopted to compute LST starting from brightness temperature:

(a)　USGS Landsat 4 Collection 2 Tier 1 TOA Reflectance (LANDSAT/LT04/C02/T1_TOA);
(b)　USGS Landsat 5 Collection 2 Tier 1 TOA Reflectance (LANDSAT/LT05/C02/T1_TOA);
(c)　USGS Landsat 7 Collection 2 Tier 1 TOA Reflectance (LANDSAT/LE07/C02/T1_TOA);
(d)　USGS Landsat 8 Collection 2 Tier 1 TOA Reflectance (LANDSAT/LC08/C02/T1_TOA);
(e)　USGS Landsat 9 Collection 2 Tier 1 TOA Reflectance (LANDSAT/LC09/C02/T1_TOA).

It is worth noting that, to compute LST from Landsat missions, TOA datasets bands B6 (from Landsat 4–5–7) and B10 (from Landsat 8–9) have been used.

Landsat data were processed in GEE [43] by adopting the approach proposed by [36]. Surface emissivity maps, needed for LST computation from Tb images, were obtained according to the Fractional Vegetation Cover (FVC) approach [64]. Therefore, to obtain FVC, NDVI was computed from surface-reflectance Landsat collections retrieved from all GEE collection products as follows:

(1)　USGS Landsat 4 Level 2, Collection 2, Tier 1 (LANDSAT/LT04/C02/T1_L2);
(2)　USGS Landsat 5 Level 2, Collection 2, Tier 1 (LANDSAT/LT05/C02/T1_L2);
(3)　USGS Landsat 7 Level 2, Collection 2, Tier 1 (LANDSAT/LE07/C02/T1_L2);
(4)　USGS Landsat 8 Level 2, Collection 2, Tier 1 (LANDSAT/LC08/C02/T1_L2);
(5)　USGS Landsat 9 Level 2, Collection 2, Tier 1 (LANDSAT/LC09/C02/T1_L2).

According to [36,64], FVC from NDVI and emissivity were computed as follows, respectively (collection from points 1 to 5). According to previous studies [65], NDVI was computed [66,67]:

$$\mathrm{NDVI} = \frac{\mathrm{NIR} - \mathrm{RED}}{\mathrm{NIR} + \mathrm{RED}} \tag{1}$$

Fractional vegetation cover (FVC) was computed as follows:

$$\mathrm{FVC} = \frac{\mathrm{NDVI} - \mathrm{NDVI_S}}{\mathrm{NDVI_V} - \mathrm{NDVI_S}} \tag{2}$$

where $\mathrm{NDVI_s}$ and $\mathrm{NDVI_v}$ are the NDVI values corresponding to completely bare soil and vegetated pixels, respectively [68]. It is worth noting that $\mathrm{NDVI_s}$ and $\mathrm{NDVI_v}$ were set to 0.2 and 0.86, respectively.

$$\varepsilon = \mathrm{FVC}\varepsilon v + (1 - \mathrm{FVC}\varepsilon s) \tag{3}$$

where $\varepsilon$ is the emissivity and $\mathrm{FVC}\varepsilon v$ and $\mathrm{FVC}\varepsilon s$ are the FVC values computed for a completely vegetated and a pure bare soil pixel, respectively.

Once emissivity maps were obtained for all the acquisitions, corresponding LST images were finally computed by the Statistical Mono-Window (SMW) algorithm from the Climate Monitoring Satellite Application Facility (CM-SAF). This technique uses an empirical relationship between $T_b$ and LST [34], based on a linearization of the radiative transfer equation showing an explicit dependence from emissivity.

$$\mathrm{LST} = A_i \left( \frac{T_b}{\epsilon} \right) + \frac{B_i}{\epsilon} + C_i \tag{4}$$

where $T_b$ is the TOA brightness temperature, and $\varepsilon$ is the surface emissivity. $A_i$, $B_i$, and $C_i$ are coefficients modelling the Total Column Water Vapor (TCWV) effect on LST. These coefficients have been made available by the NCEP/NCAR re-analysis 1 project and can be accessed and used through GEE depending on the considered Landsat collection.

Landsat data were analyzed from 1984 to 2022 including, therefore, 39 years of Landsat data. All acquisitions have been considered with more than 900 images and bidirectional reflection disturbance compensated with a self-made function in GEE according to [69] regarding the NDVI. Clouds, shadows, and saturated pixels have been masked out by considering pixel quality and radiometric saturation layers for each scene. Since the merged Landsat collections were not equally distributed in time and are therefore not suitable to perform timeseries analysis due to temporal gaps. All data have been filtered with a Savitzky–Golay filter [70–72] and regularized with a monthly timestep on GEE by adopting the Open Earth Engine Library (OEEL). It is worth noting that the year 2012 was derived after creating yearly composites through linear interpolation as explained below.

Landsat images after November 2011 (last acquisition by Landsat 5) and before March 2013 (beginning of Landsat 8 mission) were retrieved by interpolation and regularization due to the lack of images during the period mentioned above. Landsat 7 data starting from 31 May 2003 onwards were not considered and therefore not included in the timeseries regularization phase due to the failure of the Scan Line Corrector (SLC) which has affected the usage of these images. The correction of SLC was not performed with the ENVI tool, despite there being an algorithm able to do it, because we processed them in GEE.

Then, yearly composite images were computed for each year in the time range 1984–2022 by using the ee.Reducer GEE function in order to obtain the mean and the maximum pixel values in each year.

It is worth noting that LST is normally and more accurately estimated by using nighttime-acquired images to avoid the effect of direct solar irradiation in case of study of UHI. Nevertheless, the present study has focused on LST maximum trends that normally occur during the day. Moreover, the risk and exposure to the population are higher during sunlight. For these reasons, this research has been focused on daytime LST.

From the LST stack, computed and regularized maximum and mean trends were modelled in [73] with a 1st-order polynomial and the significance of the related gain evaluated performing Pettitt's trend test in R Studio [62,74–76]. It is worth noting that only significant gain values were averaged at the class level (distribution of population in each municipality in the study area). Finally gain, offset, standard errors, and *p*-value maps were realized in order to join these data with the Meta Population dataset to assess population risk from rising temperatures.

### 2.3. HDX Meta Population Dataset

The HDX Meta Facebook Population dataset was obtained as follows. Under the assumption that buildings act as a proxy for where people live, Meta (previously known as Facebook) obtains population estimates on a country-wide level, with $1 \times 1$ arcsecond resolution (~$30 \times 30$ m at the equator) and sensitivity to individual buildings, enabling accurate studies of population aggregation in rural areas. To enable global analysis, Meta has developed a building-detection model. The Meta pipeline consists of several steps: extraction of $64 \times 64$-pixel images (patches) around detected straight lines using a conventional edge detector, which reduces the amount of data for classification by a factor of approximately 4. A portion of those candidates are sampled across all countries and labeled as training and evaluation data for the CNNs. The computer vision models are trained on a single machine with four GPUs, whereas the classification runs on Meta Facebook's infrastructure on a CPU cluster. During this phase, three different types of CNN were used: a classification model based on the SegNet [50]; a feedback neural network (FeedbackNet) performing weakly-supervised segmentation of the satellite images enabling Facebook to obtain building footprints [50]; and a denoising network which is capable of improving the quality of the source data by removing high-frequency noise from the satellite imagery. The encoder–decoder-style SegNet is customized to perform the classification at the level of a patch. The encoder (a convolutional sub-network) is used to extract abstract information about the input, and the decoder (a deconvolutional sub-network) is trained to upsample the output of the encoder into a spatially meaningful probability map representing the

possibility of house existence in the input. The probabilities generated by the decoder are averaged over all spatial locations within the patch to derive the final classification, including GNSS tracking. This yields high accuracy and a reduced false positive rate on a global scale compared to other methods. To facilitate a generalized and scalable model, Meta employs weakly-supervised learning that takes the abundant and easy-to-acquire image-level categorical supervision (binary labeling) into training, and performs pixel-level prediction during deployment [50]. The methodology is motivated by the feedback mechanism in human cognition and recent advances of computational models in Feedback Neural Networks [50], which deactivates the non-relevant neurons within hidden layers of neural networks and achieves pixel-wise semantic segmentation. Both models are trained on 150,000 binary-labeled (building/no building) patches, randomly sampled from all countries and seasons, covering both rural areas and urban areas. The output layer was validated considering censuses at a country level, reaching a global overall accuracy of 98.3% [50]. Then these data have been yearly coupled with aggregated tracking from Meta phone applications (such as Instagram, Facebook, WhatsApp). These data can be accessed through Meta Data for Good (https://dataforgood.facebook.com/, last accessed on 19 April 2023). The format is raster (30 m GSD) or a dataframe, and the updating frequency is yearly or more under request. In each dataset the pixel value reports the population number according to a given characteristic.

To test the quality of the Meta population 2020 product in a rural and mountain area such as Aosta Valley Autonomous Region, in the northwest of Italy, this dataset was tested considering the 2020 census at municipality level in Aosta Valley. The HDX Meta Population dataset was considered as the predicted population while the regional census was the observed true population. For each municipality in Aosta Valley the Mean Absolute Error (MAE) was computed as follows:

$$\text{MAE} = \frac{\sum_{i=1}^{n}(p_i - o_i)}{n} \tag{5}$$

where $p_i$ is the prediction (Meta Population), $o_i$ is the observed true value (ISTAT Population), and n is the number of samples (in this case the number of Aosta Valley municipalities 74) see Table A1 in Appendix A.

The population dataset, properly processed, allowed the spatial distribution of the following variables to be obtained according to the international standard of the World Bank (see Table 1) and adopted into the present study.

**Table 1.** HDX Meta Population dataset properly processed structure.

| Population Structure | Description |
|---|---|
| VDA general | Overall population within a pixel |
| VDA men | Male population within a pixel |
| VDA women | Women population within a pixel |
| VDA women of reproductive age 15–49 | Women population in reproductive age between 15–49 years old |
| VDA elderly 60 plus | Population older than 60 years old |
| VDA children under 5 | Children population younger than 5 years old |
| VDA youth 15–24 | Young population aged between 15 and 24 years old |

### 2.4. Other Geospatial Layers

Since LST trends can reasonably be affected by multiple factors, some of them were considered and a correlation was tested in order to decide if it was reasonable to develop a multivariate suitability model including all of them or not. A multiple-correlation analysis in R was performed considering population distribution according to the following parameters: (a) fractional vegetation cover (FVC), (b) altitude, (c) slope, (d) aspect,

(e) potential incoming solar radiation (mean sunlight duration in the meteorological summer season), (f) LST gain maximum, and (g) LST gain mean.

### 2.4.1. Fractional Vegetation Cover

Fractional vegetation cover was computed from Landsat data to calibrate the LST as previously reported. Moreover, to define the vegetation percentage in a single present layer used as possible input into the risk model, FVC was also estimated in ESA SNAP 8.0.0 open-source software, starting from Copernicus Sentinel-2A (S2A) images. In particular a mean composite multi-band image in the 2020 summer meteorological season (from 1 June to 31 August) was generated in GEE with the function .mean() after applying cloud and shadow masking, and the bidirectional reflectance distribution function (BRDF) with a self-made algorithm implemented in GEE. The composite output was exported from GEE, preserving the native resolution of each S2A band and then processed in ESA SNAP 8.0.0 by applying the Biophysical Processor S2_10 m function. FVC was considered in the model in order to assess if vegetation may have a mitigating effect on temperature trends [8].

### 2.4.2. Potential Incoming Solar Radiation and Terrain Analysis

As previously said, the population distribution was analyzed considering also the geomorphology and sun irradiance. In particular, altitude and duration of solar insolation have been considered. In order to retrieve these two parameters, the Aosta Valley Digital Terrain Model (DTM) and Digital Surface Model (DSM) retrieved from aerial Lidar flight during 2005/2008 were adopted. These datasets, with a native spatial resolution of 2 m, were oversampled at 30 m with a bilinear interpolation, as described in [73]. The DTM was used to map the altitude while the DSM was used to map the duration of insolation at pixel level as a mean of the entire 2020 meteorological summer season. In Table 2 the settings parameters adopted are reported:

**Table 2.** Modelling solar duration.

| | |
|---|---|
| Solar Constant ($Wm^{-2}$) | 1367 |
| Time Period | Range of days |
| Start day | 1 June 2020 |
| End day | 31 August 2020 |
| Resolution (day) | 1 |
| Time Span (h) | 24 |
| Resolution (h) | 0.5 |
| Atmospheric Effects | Lumped Atmospheric Trasmittance |

To detect surface height, the regional deep learning dataset realized in 2020 was adopted. In fact, this dataset contains the buildings patches, with their heights, on the Aosta Valley territory. Slope and aspect were computed from the Aosta Valley Digital Terrain Model (DTM) freely available in the SCT Geoportale della Regione Autonoma Valle d'Aosta (https://geoportale.regione.vda.it/, last accessed on 30 January 2023) in SAGA GIS v.8.5.0.

### 2.5. Geostatistical Analysis

Before performing geostatistics and modeling LST trends, a normal distribution test was executed to understand the type of data. In particular, a Kolmogorov–Smirnov test concerning LST profile time-series was performed in R. Then, a self-made script in R Studio was adopted to map gain, offset, and *p*-value (Pettitt's test). Pettitt's test was carried out because, as indicated in the workflow, only the significant pixels were modeled. Therefore, thanks to it, break points were identified in the time series. In LST max and mean gain layers, pixels not significant were masked out considering only significant Pettitt's $p$-value $< 0.05$ (hereinafter called CM) after clipping the data onto the Population dataset. Subsequently, the ISODATA unsupervised classification–clustering algorithm was performed on CM and

the separability of each class obtained was checked by computing the Jeffries–Matusita (J-M) distances as follows:

$$JM_{ub} = \sqrt{2(1 - e^{-\alpha})} \tag{6}$$

$$\alpha = \frac{1}{8}(\mu_u - \mu_b)^T \left(\frac{C_u + C_b}{2}\right)^{-1} (\mu_u - \mu_b) + \frac{1}{2}\ln\left[\frac{\frac{1}{2}|C_u + C_b|}{\sqrt{|C_u| \times |C_b|}}\right] \tag{7}$$

where:

u and b: the classes to separate,
$C_u$: the covariance matrix of u,
$\mu_u$: the mean vector of u,
T: transposition function.

The same procedure was applied to the FVC, solar duration, quote, aspect, and slope. Each cluster (hereinafter called CLU) generated was considered in the model only if it had a strong and statistically significant correlation to the other to assess population exposure by performing zonal statistics in SAGA GIS v.8.5 [73]. All the input data have been normalized. The maximum number of iterations has been set to 20 while the initial number of clusters and samples in the cluster is 5 and the maximum number of clusters is 16. Finally, in each CLU, zonal statistics considering Aosta Valley municipalities were performed. In particular, to assess rising temperature exposure, the following procedure was deployed:

$$Rexp = CLU_{LST_{gain}}(x, y) \cap Meta_{pop}(x, y) \tag{8}$$

where:

$R_{exp}$ is the risk exposure;
$CLU_{LSTgain}$ is the Cluster performed on LST maximum and mean significant layer respectively;
$Meta_{pop}$ is the Meta population processed dataset.

Since in each municipality different clusters of the same type were present after performing Equation (8) to define the risk exposure in each municipality, considering each type of cluster, a pivot spatial table was realized by using the Group Stat tool available in QGIS.

Moreover, a suitability risk map modeler was realized and performed starting from native input datasets to map rising temperature risk exposure. An analytic hierarchical process (AHP) was followed, and it was decided that only correlated and significant variables would be tested after performing a multivariate geostatistical analysis (see in Section 3). The risk was mapped as follows:

$$R_{exp} = \{[(Z_\alpha(x, y) \times \omega_\alpha) + (Z_\beta(x, y) \times \omega_\beta) + (Z_\gamma(x, y) \times \omega_\gamma)] \times 100\} \tag{9}$$

$$\omega = \frac{M}{(n + M - 1)} \tag{10}$$

where $R_{exp}$ = risk exposure.

$$TfMSSmallZ = \frac{\sigma \times \vartheta}{[X - (\tau \times \mu) + (\sigma \times \vartheta)]}$$

where:

$\sigma$ = standard deviation;
$\mu$ = mean;
$\vartheta$ = a $\sigma$ multiplier;
$\tau$ = a $\mu$ multiplier;
$\omega$ = input value;

ω = the weight defined in each input (in this case 0.333);
M = the maximum of the AHP scale;
n = the number of criteria (in this case 10);
α, β, γ = the three input datasets respectively (LST Gain mean, LST Gain max, and DSM).

## 3. Result

The first phase was to check the consistency of the Meta Population dataset. Therefore, the MAE was computed as reported in Equation (5). The predictor was assumed to be the Meta Population dataset and the true value data to be the ISTAT Population in each Aosta Valley municipality. In Appendix A (in Table A1) we report the MAE results obtained. The dataset seems to have a consistency by observing the results obtained. Moreover, the gender and age distribution were tested but not reported showing a strong consistency. After checking the quality of the population dataset and modeling the LST maximum and mean trend and defining their significance, correlations were tested considering the following parameters: (a) fractional vegetation cover (FVC), (b) quote, (c) slope, (d) aspect, and (e) potential incoming solar radiation (mean sunlight duration in the meteorological summer season). In Figure 2 we report the results obtained. LST gain mean and maximum are positively and strongly correlated with each other, as computed by their linear R Pearson coefficient, as well as slope (but with a negative correlation). Moreover, they are statistically significant, with $p$-value < 0.05. The other variables have a lower Pearson correlation coefficient and are not statistically significant. Therefore, we decided to model population distribution in each variable but computed the risk exposure model in two ways: the first as reported in Equation (8) considering only the LST gain and clustering them, and then computing and mapping each Aosta Valley municipality, taking into account the population involved according to their structural parameters (please see Table 1); the second way was by mapping, with a suitability model, the population risk exposure as reported in Equation (9).

In the tables below (see Table 3), the ISODATA clustering performed per LST Gain Max has been reported, which represents the major risk exposure in a given area with the optimal number of clusters with their mean range, standard deviation, and distance. It is worth noting that, onto these clusters, Equation (8) was computed, a map generated, and zonal statistics performed by realizing a pivot spatial analysis onto population datasets at municipality level. In this work only LST Gain Max tables have been reported due to the fact that they represent the extreme conditions.

**Table 3.** LST Clusters.

| Cluster ID | Mean Gain LST Max (°C) | StDev Gain LST Max | Exposure–Risk Class Assessment | Mean Distance |
|---|---|---|---|---|
| 1—I | 0.21 | 0.00 | 7 | 0.05 |
| 2—II | 0.01 | 0.04 | 1 | 0.28 |
| 3—III | 0.19 | 0.01 | 6 | 0.06 |
| 4—IV | 0.27 | 0.02 | 9 | 0.10 |
| 5—V | 0.23 | 0.01 | 8 | 0.08 |
| 6—VI | 0.17 | 0.01 | 5 | 0.08 |
| 7—VII | 0.10 | 0.02 | 3 | 0.07 |
| 8—VIII | 0.31 | 0.04 | 10 | 0.16 |
| 9—IX | 0.07 | 0.00 | 2 | 0.13 |
| 10—X | 0.13 | 0.00 | 4 | 0.09 |
| 11—XI | 0.38 | 0.00 | 11 | 0.29 |

In the figure below, the maps obtained from Equation (8) and from which Table A2 in Appendix B was obtained have been reported. The exposure-risk class assessment reported in the map was defined considering the gain values, from major to minor, where higher gain values have higher class numbers. Consider Figures 3 and 4.

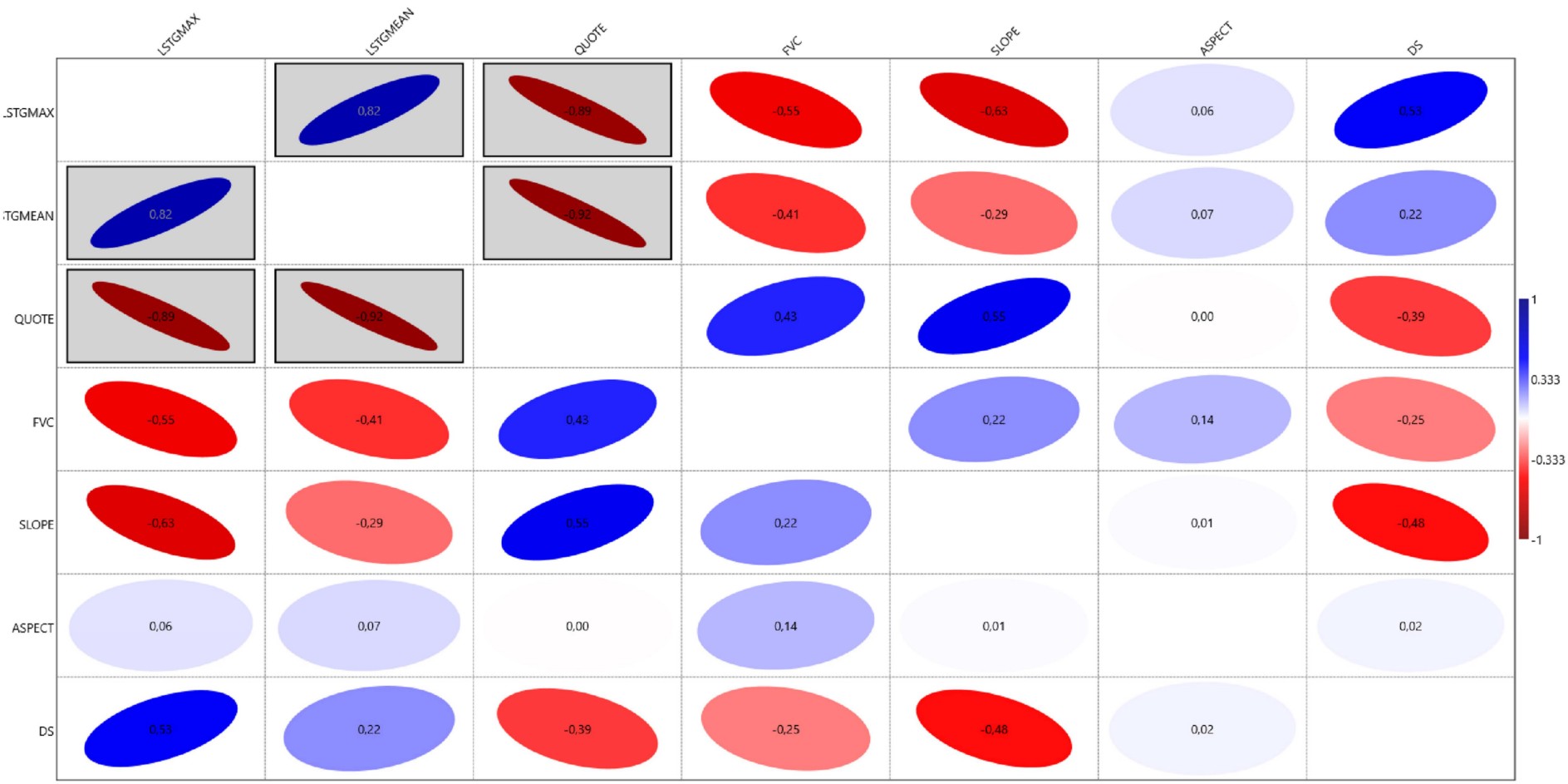

**Figure 2.** Multiple correlation testing involving LST Gain Maximum, LST Gain Mean, Quote, FVC (Fractional Vegetation Cover), Slope, Aspect, Sun Duration (DS). Positive correlation is in blue, negative in red. Rectangles represent statistically significant *p*-value < 0.05.

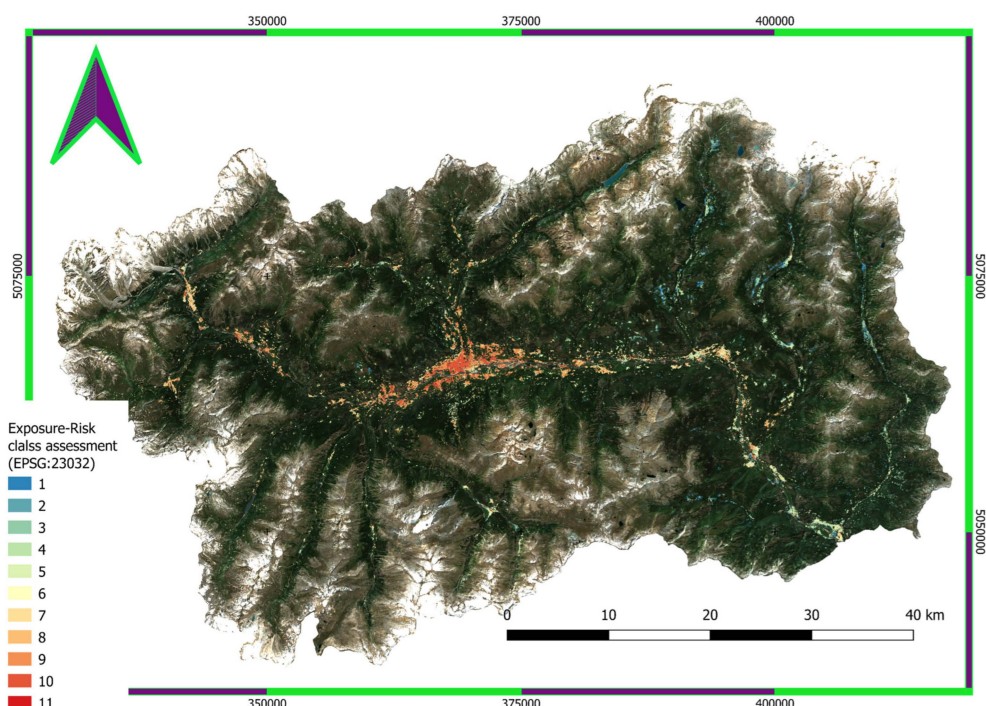

**Figure 3.** Exposure-Risk class assessment map (scale 1:3,300,000). EPSG: 23032.

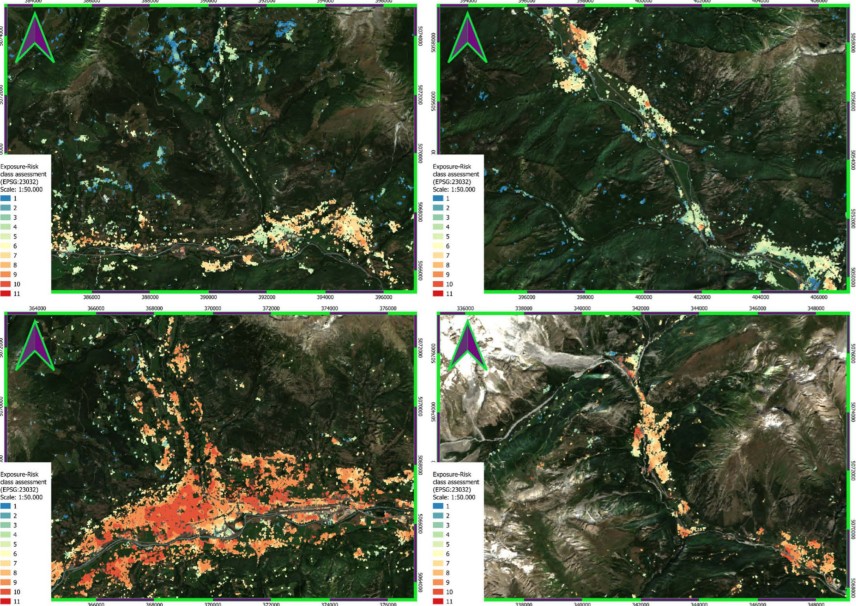

**Figure 4.** Exposure-Risk class assessment map (with some zooms in in the bottom valley from East to West with a scale of 1:50,000). EPSG: 23032.

Then, as reported previously, a risk map was developed by adopting Equation (9). In this last case the map produced from the suitability risk modeler has considered not only LST gain maximum such as those in Figures 3 and 4 but also LST gain mean and quote, due to the fact that they have been tested as previously described. It is worth noting that the model developed suggests a risk exposure to rising LST, taking into consideration the variables previously described in risk-exposure ranges between 0 and 100%, where 100% is the maximum risk exposure considered according to the input parameters considered in the model. In these areas, attention must be paid to the high risk represented by the LST trends and their locations. In these areas, the emissivity of the materials, and consequently their

albedo, is very different from those with materials that are able to mitigate heatwaves and LST trends. The results obtained have permitted the population involved to be mapped, thanks to the zonal statistics performed adopting the HDX Meta population dataset.

From these last maps obtained from the risk modeler developed, it is possible to see how more than 60% of the Aosta Valley population, who mostly live in the bottom of the valley, are exposed to a risk of rising temperature and heatwaves with a probability greater than 55%. These effects have and will have important socio-economic impacts, not only on the health sector but also on pets and domestic animals (particularly if we consider that one of the main items of the GDP of Aosta Valley comes from animal husbandry and how, therefore, certain breeding farms and related production and animal welfare are more at risk than others, although the practice of summer pasture can mitigate this through a mechanism of escape from mapped and modeled thermal trends) (please see Figures 5 and 6).

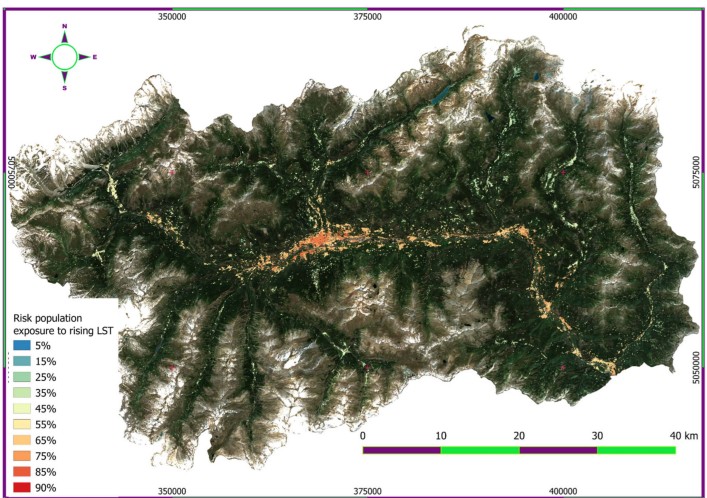

**Figure 5.** Risk population exposure to rising LST mapped according to Equation (9). EPSG: 23032.

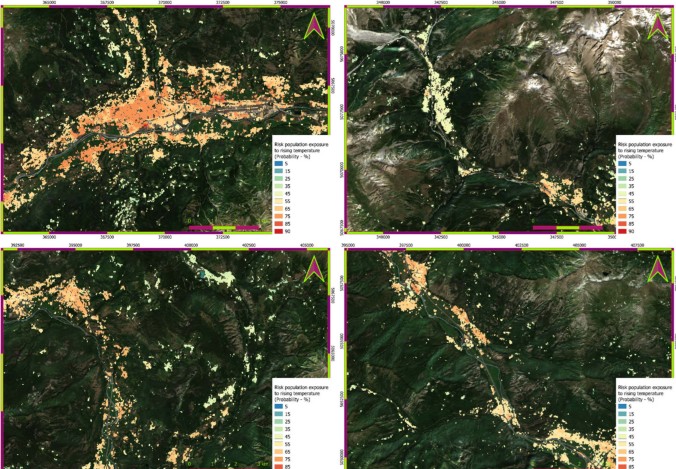

**Figure 6.** Risk population exposure to rising LST mapped according to Equation (9) (with zooms in given areas with a scale of 1:50,000), EPSG: 23032.

## 4. Discussion

The results obtained constitute a first example on the Aosta Valley territory of climate risk assessment through the use of a unique high-resolution dataset (being currently the most detailed raster data for scientific applications). Moreover, the Meta dataset has never been used until today, based on the information available in the scientific literature, in this

way. A combined use with Earth observation data or weather station data can certainly be an important tool of added value in the creation of new information layers and GIS and risk analysis. The maps obtained can thus represent added value in the context of upgrading hospitals and clinical outpatient clinics, or even from an urban planning perspective with regard to urban-greening policies and actions. We hope this instrument will have added value for civil defense. Certainly, a future development that focuses on individual buildings taking into consideration increasingly timely data of population and physical characteristics of surfaces is desirable so as to provide more and more detailed, but most importantly up-to-date, data.

In this study, it is worth noting that the HDX Meta population dataset from empirical ground data collected by some municipalities in Aosta Valley seemed to be more accurate in terms of people that are present in a given area than the ISTAT residents. In fact, many people are resident in a given municipality but live in a different municipality in which they are domiciled. Therefore, a population dataset based on a tracking system seems to better fit the real population distribution which still remains a great challenge considering also privacy policies.

Given the quality of the data, an increasingly up-to-date and detailed population dataset at both spatial and temporal resolution levels is desirable for studies of this kind. Moreover, cross-referencing such data with geo-referenced animal population data would certainly allow the development of models of the risk to domestic animals and their productivity in the case of non-pet animals, while also allowing the development of new lines on animal welfare with a view to adapting to climate change while avoiding stresses such triggering possible disease from a weakened situation.

Nowadays, in fact, there is no high-detail geospatial dataset of both domestic and affectionate animal populations (such as cats and dogs) at a global level. A datum of this kind would allow not only a modeling of climate risk but also, and above all, health risk in the case of zoonosis and eco-epidemiology towards a real and concrete application of One Health. It is worth noting that the implementation would not be complex, considering that in many countries there is an obligation to chip pets and in the case of livestock there is a specific registry managed by veterinary services.

Surely a global effort in this direction together with a high temporal resolution and good detail mapping of the population with information aggregated in respect of privacy would allow a significant technological transfer to the health sector through cross studies for health and risk analysis related to the effects of climate change. In this case, an application on a local scale was attempted, also testing the quality of the population dataset. Surely, future missions such as those from the Albedo enterprise which will provide thermal data with GSD 2 m (if they will be made free for applied scientific research) will allow extremely detailed studies if they are accompanied by other very-high-resolution information datasets. With the data currently in possession, however, precious information can be obtained from a planning and management point of view. It still seems difficult and complex to suggest punctual actions at the sub-district level. In fact, it must be said that the Landsat data have a thermal resolution at 30 m, equal to 900 square meters but resampled by NASA and that the thermal sensors have an average resolution of 100 m, equal to an area of one hectare.

Currently the only scientific mission with a higher-resolution thermal sensor, ECOSTRESS on board the International Space Station, does not allow long-term studies and was mainly designed as a tool for monitoring vegetation. Unfortunately, other scientific missions that make satellite data available free of charge, such as the European space program's Copernicus, have thermal data that are not suitable for conducting detailed studies. In fact, Sentinel-3 has a GSD of 1 Km. The development of high thermal resolution sensors with free access data would be desirable. Only the Albedo company is currently investing in high-resolution commercial satellite data as previously said, but it is not yet known whether its distribution policies will be free for research. However, in a mitigation and adaptation perspective to climate change, their implementation is not only desirable on a global level but also strategic in defining concrete One Health actions [77–79]. It is worth noting that

mapping the exposure of population involved according to different levels of temperature (LST) gain permits greening actions and policies to be addressed, favors the identification of new medical or health centers, predicts in advance the areas that will be more subject to emergency calls, redevelops areas or zones most at risk with a view to mitigation and above all adaptation, makes forecasts on access to hospitals in case of heat waves and the impact of costs on the health sector having mapped data, and evaluates the effectiveness of requalification policies and actions and its effects on heat flows and on the risk associated with the exposed population. Nevertheless, at the present time, analyses of exposures to thermal trends are linked to EO data with GSD at 30 m (natively 100 m resampled at 30 m in case of Landsat). They currently represent the highest resolution available for scientific purposes. The hope is that the missions of the private company Albedo, which will provide satellite thermal data at 2 m GSD, will be free for scientific purposes and will allow a significant technological and application leap. An increasingly detailed population dataset is also desirable, although the aggregate Meta dataset is currently the most detailed from a spatial point of view. To date, in fact, although the present application is notable, it still limits the analyses at a cluster level by areas, making analyses at a building level more complex, which would certainly be desirable for the future. In fact, mapping the risk at the level of a single residential structure and its surroundings would allow increasingly precise actions with a view to adaptation and capillary and punctual analysis of the risk.

Concerning the present study, it is interesting to see how the bottom of the valley is more affected by rising temperature and how FVCs do not play a statistically significant role (probably this is due to the fact that Landsat pixel GSD does not permit one to appreciate in urban areas the effect offered by sparse vegetation (that normally, considering the study area, is less than half a pixel). It is interesting to know how most of the highly risk areas are located in industrial areas and in modern buildings rather than in historical buildings. At the same time, it is interesting to underline from a civil protection perspective how more than 60% of the Aosta Valley population (mostly concentrated in the valley floor for work reasons) is in high exposure and risk classes with both approaches adopted. Although in the summer some prefer to find refuge in the side valleys, the fact that a large part of the mostly elderly population is exposed to a greater risk should lead to rethinking urban planning and the creation of services or assistance hubs in areas with greater exposure. We hope to see a major application of EO Data in a One Health perspective [80].

Regardless of the considerations on the planning developments of climate adaptation and mitigation, we hope that this study will be useful and can also be scaled to other realities and become more and more detailed.

## 5. Conclusions

The rising temperatures due to the effects of climate change require the rapid development of adaptation and mitigation plans and concrete actions. In this study, an attempt was made for the first time by coupling the HDX Meta population dataset and free satellite data from the USGS NASA Landsat 4–9 missions. In particular LST trends were used to map the exposure and risk deriving from the increase in temperatures in Aosta Valley, the smallest region of Italy but one of the hardest hit by the effects of climate change. The developed approaches can be scaled to other realities and at national and international level by enhancing not only the Meta data but also by promoting a technological and knowledge transfer to the health and environmental sector through concrete tools useful for tackling climate change. In the case of the Aosta Valley, the punctual mapping of the risk and trend of exposure to temperatures has made it possible, thanks to the Meta dataset, to define risk and exposure classes according to the distribution and structure of the population at the municipal level which we hope will be useful to the sector of civil protection and medical-health including veterinary. If we think of domestic and bred animals, their spatial distribution has not been considered. Therefore, further studies on them as well as more and more punctual monitoring of the population with increasingly updated data are

certainly desirable in the climate change framework adaptation and mitigation planning, adaptation and policies.

Finally, results have shown a steeper increase in LST maximum trend, especially in the bottom valley municipalities, and especially in new built-up areas around factories where more of 60% of the Aosta Valley population (especially elderly and younger people) live and where a high exposure risk has been detected and mapped with both approaches performed. We strongly hope that maps produced may help the local planners and the civil protection services to face global warming in a One Health perspective. Last but not least, we hope this type of application may become ordinary and useful to other regions, countries, studies, and more in general realities enhancing the exploitation of a combined use of free satellite data and population data social tracking for the purposes of rational territorial planning and management according to a real One Health approach.

**Author Contributions:** Conceptualization, T.O. and A.V.; methodology T.O. and A.V.; software, T.O.; validation T.O., D.C. and E.B.-M.; formal analysis, T.O.; investigation, T.O. and A.V.; resources, T.O.; data curation, T.O., A.V. and B.M.; writing—original draft preparation, T.O. and A.V.; writing—review and editing, T.O. and A.V.; visualization, T.O. and A.V.; supervision, T.O., A.V., D.C. and E.B.-M.; project administration, T.O. All authors have read and agreed to the published version of the manuscript.

**Funding:** This research received no external funding.

**Data Availability Statement:** The maps obtained, and all the data used, in this research are available through request by e-mail to the corresponding author.

**Acknowledgments:** A remarkable thanks to GEO4Agri DISAFA Lab colleagues and Andreas Gros as well as to HRSL Core Science Meta Research of Meta Facebook as well as IN.VA spa GIS Unit and eoVdA.

**Conflicts of Interest:** The authors declare no conflict of interest.

## Appendix A

Below, the MAE computed considering the HDX Meta Population dataset and the ISTAT population data is reported:

**Table A1.** Population validation.

| ID Italian Municipality | Municipalities in Aosta Valley Region | HDX Meta Population 2020 ($p_i$) | ISTAT Effective Resident on 31 December 2020 ($o_i$) | MAE |
|---|---|---|---|---|
| A205 | Allein | 244 | 210 | 34 |
| A305 | Antey-Saint-Andre | 645 | 565 | 80 |
| A326 | Aosta | 33,204 | 33,916 | −712 |
| A424 | Arnad | 1278 | 1245 | 33 |
| A452 | Arvier | 917 | 870 | 47 |
| A521 | Avise | 378 | 306 | 72 |
| A094 | Ayas | 1406 | 1393 | 13 |
| A108 | Aymavilles | 2266 | 2104 | 162 |
| A643 | Bard | 110 | 122 | −12 |
| A877 | Bionaz | 224 | 225 | −1 |
| B192 | Brissogne | 1034 | 948 | 86 |
| B230 | Brusson | 792 | 883 | −91 |
| C593 | Challand-Saint-Anselme | 804 | 758 | 46 |
| C594 | Challand-Saint-Victor | 618 | 548 | 70 |
| C595 | Chambave | 908 | 919 | −11 |
| B491 | Chamois | 92 | 98 | −6 |

**Table A1.** *Cont.*

| ID Italian Municipality | Municipalities in Aosta Valley Region | HDX Meta Population 2020 ($p_i$) | ISTAT Effective Resident on 31 December 2020 ($o_i$) | MAE |
|---|---|---|---|---|
| C596 | Champdepraz | 742 | 714 | 28 |
| B540 | Champorcher | 365 | 394 | −29 |
| C598 | Charvensod | 2688 | 2338 | 350 |
| C294 | Chatillon | 5023 | 4524 | 499 |
| C821 | Cogne | 1403 | 1377 | 26 |
| D012 | Courmayeur | 2759 | 2761 | −2 |
| D338 | Donnas | 2551 | 2448 | 103 |
| D356 | Doues | 580 | 512 | 68 |
| D402 | Emarese | 247 | 223 | 24 |
| D444 | Etroubles | 539 | 481 | 58 |
| D537 | Fenis | 1860 | 1769 | 91 |
| D666 | Fontainemore | 472 | 431 | 41 |
| D839 | Gaby | 496 | 460 | 36 |
| E029 | Gignod | 2094 | 1715 | 379 |
| E165 | Gressan | 3819 | 3393 | 426 |
| E167 | Gressoney-La-Trinite | 315 | 318 | −3 |
| E168 | Gressoney-Saint-Jean | 815 | 812 | 3 |
| E273 | Hone | 1170 | 1189 | −19 |
| E306 | Introd | 696 | 661 | 35 |
| E369 | Issime | 430 | 407 | 23 |
| E371 | Issogne | 1405 | 1349 | 56 |
| E391 | Jovencan | 895 | 723 | 172 |
| A308 | La Magdeleine | 129 | 109 | 20 |
| E458 | La Salle | 2200 | 2001 | 199 |
| E470 | La Thuile | 810 | 812 | −2 |
| E587 | Lillianes | 444 | 445 | −1 |
| F367 | Montjovet | 1864 | 1802 | 62 |
| F726 | Morgex | 2166 | 2096 | 70 |
| F987 | Nus | 3228 | 2950 | 278 |
| G045 | Ollomont | 150 | 165 | −15 |
| G012 | Oyace | 223 | 217 | 6 |
| G459 | Perloz | 413 | 457 | −44 |
| G794 | Pollein | 1617 | 1536 | 81 |
| G854 | Pontboset | 185 | 173 | 12 |
| G545 | Pontey | 907 | 801 | 106 |
| G860 | Pont-Saint-Martin | 4014 | 3592 | 422 |
| H042 | Pre-Saint-Didier | 1031 | 1031 | |
| H110 | Quart | 4601 | 4045 | 556 |
| H262 | Rhemes-Notre-Dame | 111 | 85 | 26 |
| H263 | Rhemes-Saint-Georges | 192 | 174 | 18 |
| H497 | Roisan | 1217 | 1038 | 179 |
| H669 | Saint-Christophe | 3598 | 3446 | 152 |
| H670 | Saint-Denis | 399 | 382 | 17 |
| H671 | Saint-Marcel | 1385 | 1365 | 20 |
| H672 | Saint-Nicolas | 311 | 320 | −9 |
| H673 | Saint-Oyen | 240 | 199 | 41 |
| H674 | Saint-Pierre | 3512 | 3195 | 317 |
| H675 | Saint-Rhemy | 328 | 329 | −1 |
| H676 | Saint-Vincent | 4509 | 4432 | 77 |
| I442 | Sarre | 5497 | 4817 | 680 |
| L217 | Torgnon | 525 | 567 | −42 |
| L582 | Valgrisenche | 198 | 196 | 2 |
| L643 | Valpelline | 691 | 618 | 73 |
| L647 | Valsavarenche | 187 | 175 | 12 |
| L654 | Valtournenche | 2037 | 2255 | −218 |
| L783 | Verrayes | 1389 | 1264 | 125 |
| C282 | Verres | 2712 | 2577 | 135 |
| L981 | Villeneuve | 1380 | 1259 | 121 |
| | Aosta Valley Region | 130,683 | 125,034 | 76 |

## Appendix B

Below we report the number of people and their relative structure exposed to each class of rising temperature (LST) in each Aosta Valley municipality:

**Table A2.** Risk assessment defined by adopting the first method (see Equation (8)).

| ID Italian VDA Municipality | Cluster ID—Mean Gain LST Max—n° Population Exposed VDA General | | | | | | | | | | |
| --- | --- | --- | --- | --- | --- | --- | --- | --- | --- | --- | --- |
| | I | II | III | IV | V | VI | VII | VIII | IX | X | XI |
| A205 | 34 | | 65 | | 17 | 86 | 6 | | 3 | 33 | |
| A305 | 4 | 16 | 75 | 1 | | 231 | 65 | | 38 | 215 | |
| A326 | 166 | 4 | 138 | 2805 | 546 | 66 | 7 | 12,346 | 13 | 45 | 17,069 |
| A424 | 326 | 3 | 436 | 17 | 94 | 155 | 69 | 3 | 57 | 117 | |
| A452 | 12 | | 9 | 534 | 171 | 4 | | 186 | | 1 | |
| A521 | 64 | | 32 | 95 | 143 | 10 | 21 | 2 | 1 | 10 | |
| A094 | 32 | 58 | 103 | 2 | 4 | 160 | 302 | 1 | 396 | 350 | |
| A108 | 32 | | 26 | 529 | 122 | 11 | | 1203 | | 3 | 340 |
| A643 | 14 | | 23 | | | 29 | 17 | | | 29 | |
| A877 | 17 | 1 | 55 | 23 | 31 | 47 | 8 | 5 | 17 | 20 | |
| B192 | 44 | | 54 | 427 | 178 | 39 | 1 | 281 | | 2 | 8 |
| B230 | 7 | 6 | 144 | | 1 | 304 | 58 | | 61 | 211 | |
| C593 | 184 | | 219 | 11 | 127 | 176 | 13 | | 4 | 71 | |
| C594 | 36 | | 13 | 295 | 143 | 9 | | 119 | | 3 | |
| C595 | 151 | 1 | 232 | 128 | 195 | 115 | 2 | 65 | | 18 | |
| B491 | | 3 | | | | 1 | 27 | | 29 | 32 | |
| C596 | 214 | 1 | 217 | 6 | 114 | 164 | 4 | | 3 | 18 | |
| B540 | | 58 | | | | 2 | 76 | | 120 | 109 | |
| C598 | 68 | | 70 | 609 | 170 | 32 | | 1476 | | 1 | 263 |
| C294 | 785 | 6 | 1356 | 611 | 1352 | 843 | 4 | 32 | 4 | 31 | |
| C821 | 235 | 11 | 275 | 41 | 347 | 249 | 65 | 5 | 31 | 144 | |
| D012 | 224 | | 161 | 1036 | 944 | 76 | | 179 | | 3 | 135 |
| D338 | 69 | | 788 | 1 | 1 | 1289 | 32 | | 3 | 367 | |
| D356 | 39 | | 41 | 233 | 104 | 28 | 2 | 123 | | 8 | 1 |
| D402 | 33 | 2 | 62 | | | 49 | 3 | | 5 | 93 | |
| D444 | 114 | 1 | 97 | 163 | 141 | 18 | | | 1 | 3 | |
| D537 | 103 | | 28 | 986 | 604 | 15 | | 122 | | 1 | |
| D666 | 38 | 9 | 105 | | 3 | 193 | 24 | | 18 | 82 | |
| D839 | | 4 | 84 | | | 238 | 40 | | 8 | 122 | |
| E029 | 162 | 3 | 116 | 492 | 364 | 127 | 22 | 661 | 24 | 96 | 29 |
| E165 | 53 | | 64 | 230 | 165 | 70 | 1 | 1723 | | 25 | 1488 |
| E167 | 4 | 20 | 10 | | 1 | 21 | 86 | | 129 | 43 | 1 |
| E168 | 2 | 18 | 14 | | | 141 | 202 | | 184 | 253 | |
| E273 | 48 | 9 | 291 | 24 | 28 | 492 | 58 | 16 | 22 | 183 | |
| E306 | 64 | 1 | 51 | 233 | 304 | 21 | 2 | 11 | 2 | 8 | |
| E369 | 15 | 22 | 36 | | 3 | 150 | 32 | | 55 | 116 | |
| E371 | 344 | 25 | 326 | 10 | 411 | 154 | 41 | | 31 | 63 | |
| E391 | | | | 40 | | | | 502 | | | 352 |
| A308 | | 6 | | | | 11 | 23 | | 69 | 20 | |
| E458 | 164 | | 102 | 711 | 393 | 28 | 2 | 790 | 1 | 4 | 5 |
| E470 | 34 | | 27 | 364 | 240 | 3 | | 142 | | | |
| E587 | 8 | 4 | 162 | | | 210 | 8 | | 10 | 41 | |
| F367 | 302 | | 250 | 471 | 737 | 61 | 1 | 36 | 1 | 5 | |
| F726 | 32 | | 8 | 769 | 150 | 4 | | 976 | | 3 | 225 |
| F987 | 221 | 1 | 159 | 1368 | 612 | 104 | 15 | 558 | 23 | 46 | 121 |
| G045 | 22 | | 2 | 67 | 43 | 1 | | 14 | | 1 | |
| G012 | 31 | | 15 | 71 | 83 | 19 | | | | 3 | |
| G459 | | 4 | 27 | | | 203 | 31 | | 8 | 141 | |
| G794 | 15 | | 3 | 575 | 102 | 3 | 1 | 871 | | 2 | 44 |
| G854 | | 3 | | | | 37 | 67 | | 7 | 72 | |
| G545 | 126 | | 89 | 143 | 532 | 8 | | | | 9 | |

**Table A2.** *Cont.*

| ID Italian VDA Municipality | Cluster ID—Mean Gain LST Max—n° Population Exposed VDA General | | | | | | | | | | |
|---|---|---|---|---|---|---|---|---|---|---|---|
| | **I** | **II** | **III** | **IV** | **V** | **VI** | **VII** | **VIII** | **IX** | **X** | **XI** |
| G860 | 1017 | 3 | 1525 | 165 | 507 | 691 | 27 | | 7 | 72 | |
| H042 | 86 | | 38 | 318 | 365 | 8 | 3 | 182 | 2 | 6 | 25 |
| H110 | 220 | 2 | 208 | 1912 | 692 | 102 | 17 | 1209 | 6 | 51 | 182 |
| H262 | 18 | | 18 | 1 | 7 | 32 | 2 | | | 32 | |
| H263 | 37 | | 27 | 20 | 65 | 28 | | | 3 | 11 | |
| H497 | 11 | | 1 | 670 | 92 | 1 | | 440 | | | |
| H669 | 17 | | 45 | 737 | 64 | 2 | | 2340 | | | 391 |
| H670 | 45 | 11 | 46 | 35 | 59 | 63 | 35 | 7 | 28 | 71 | |
| H671 | 69 | 1 | 79 | 651 | 226 | 28 | 4 | 308 | 3 | 10 | 6 |
| H672 | 38 | | 55 | 97 | 93 | 10 | 5 | 4 | | 8 | |
| H673 | 14 | | 59 | | 3 | 82 | 24 | | 13 | 46 | |
| H674 | 16 | 12 | 16 | 830 | 77 | 34 | 17 | 1773 | 17 | 21 | 700 |
| H675 | 29 | | 37 | 24 | 40 | 140 | 2 | 12 | | 41 | 3 |
| H676 | 801 | 3 | 403 | 884 | 2068 | 117 | 34 | 93 | 22 | 83 | |
| I442 | 178 | 1 | 128 | 1655 | 358 | 139 | 4 | 2338 | 16 | 42 | 640 |
| L217 | | 56 | 6 | | | | 28 | 141 | | 134 | 159 |
| L582 | 35 | | 15 | 49 | 87 | 4 | | 4 | | 2 | 2 |
| L643 | 94 | | 87 | 202 | 185 | 22 | 1 | 93 | | 6 | 2 |
| L647 | 51 | 1 | 21 | 19 | 46 | 29 | 1 | 2 | 1 | 16 | |
| L654 | | 81 | 73 | | | 426 | 381 | | 358 | 717 | |
| L783 | 191 | 8 | 242 | 75 | 234 | 310 | 95 | 7 | 43 | 184 | |
| C282 | 714 | 1 | 591 | 394 | 449 | 265 | 3 | 167 | 2 | 91 | 35 |
| L981 | 70 | | 40 | 475 | 198 | 9 | 1 | 553 | | 1 | 34 |
| VDA overall | 8443 | 480 | 10,420 | 23,334 | 15,635 | 9077 | 2235 | 31,980 | 2033 | 4946 | 22,101 |
| VDA overall % | 6.5 | 0.4 | 8.0 | 17.9 | 12.0 | 6.9 | 1.7 | 24.5 | 1.6 | 3.8 | 16.9 |

| ID Italian VDA Municipality | Cluster ID—Mean Gain LST Max—n° Population Exposed VDA Men | | | | | | | | | | |
|---|---|---|---|---|---|---|---|---|---|---|---|
| | **I** | **II** | **III** | **IV** | **V** | **VI** | **VII** | **VIII** | **IX** | **X** | **XI** |
| A205 | 17 | | 33 | | 6 | 46 | 3 | | 1 | 16 | |
| A305 | 2 | 9 | 34 | | | 114 | 32 | | 19 | 105 | |
| A326 | 86 | 2 | 72 | 1381 | 274 | 35 | 4 | 5852 | 7 | 23 | 7704 |
| A424 | 155 | 1 | 203 | 8 | 43 | 73 | 34 | 1 | 29 | 56 | |
| A452 | 6 | | 4 | 273 | 84 | 2 | | 96 | | 1 | |
| A521 | 31 | | 16 | 45 | 69 | 6 | 10 | 1 | 1 | 5 | |
| A094 | 17 | 31 | 54 | 1 | 2 | 80 | 154 | | 194 | 174 | |
| A108 | 16 | | 12 | 262 | 61 | 5 | | 597 | | 1 | 168 |
| A643 | 6 | | 11 | | | 13 | 8 | | | 13 | |
| A877 | 9 | | 29 | 12 | 16 | 27 | 4 | 2 | 7 | 12 | |
| B192 | 26 | | 28 | 218 | 92 | 21 | | 144 | | 1 | 4 |
| B230 | 4 | 4 | 70 | | | 142 | 29 | | 30 | 98 | |
| C593 | 90 | | 107 | 6 | 63 | 90 | 7 | | 2 | 38 | |
| C594 | 18 | | 7 | 147 | 73 | 5 | | 61 | | 2 | |
| C595 | 75 | 1 | 113 | 67 | 99 | 56 | 1 | 34 | | 10 | |
| B491 | | 2 | | | | 1 | 13 | | 14 | 15 | |
| C596 | 100 | 1 | 103 | 3 | 57 | 79 | 2 | | 2 | 9 | |
| B540 | | 27 | | | | 1 | 37 | | 59 | 56 | |
| C598 | 35 | | 37 | 303 | 86 | 17 | | 731 | | | 131 |
| C294 | 375 | 3 | 658 | 303 | 656 | 413 | 2 | 15 | 2 | 15 | |
| C821 | 115 | 5 | 135 | 20 | 177 | 123 | 33 | 2 | 15 | 70 | |
| D012 | 113 | | 84 | 510 | 468 | 40 | | 90 | | 1 | 71 |
| D338 | 33 | | 371 | | 1 | 615 | 16 | | 2 | 179 | |
| D356 | 19 | | 20 | 114 | 51 | 14 | 1 | 58 | | 4 | 1 |
| D402 | 15 | 1 | 33 | | | 26 | 2 | | 2 | 53 | |
| D444 | 58 | 1 | 49 | 83 | 71 | 9 | | | | 2 | |
| D537 | 51 | | 14 | 500 | 297 | 7 | | 63 | | 1 | |
| D666 | 20 | 5 | 53 | | 2 | 97 | 13 | | 10 | 42 | |
| D839 | | 2 | 42 | | | 115 | 21 | | 4 | 58 | |

**Table A2.** *Cont.*

| ID Italian VDA Municipality | Cluster ID—Mean Gain LST Max—n° Population Exposed VDA General | | | | | | | | | | |
|---|---|---|---|---|---|---|---|---|---|---|---|
| | I | II | III | IV | V | VI | VII | VIII | IX | X | XI |
| E029 | 78 | 1 | 60 | 233 | 175 | 68 | 12 | 326 | 12 | 50 | 15 |
| E165 | 29 | | 34 | 116 | 89 | 36 | | 859 | | 12 | 735 |
| E167 | 2 | 8 | 5 | | | 10 | 40 | | 56 | 21 | 1 |
| E168 | 1 | 8 | 6 | | | 68 | 100 | | 90 | 125 | |
| E273 | 24 | 4 | 137 | 11 | 14 | 234 | 27 | 8 | 11 | 87 | |
| E306 | 32 | | 26 | 119 | 147 | 10 | 1 | 6 | 1 | 4 | |
| E369 | 8 | 12 | 18 | | 1 | 72 | 16 | | 28 | 57 | |
| E371 | 167 | 12 | 157 | 5 | 203 | 74 | 20 | | 15 | 29 | |
| E391 | | | | 21 | | | | 255 | | | 181 |
| A308 | | 3 | | | | 6 | 13 | | 36 | 12 | |
| E458 | 82 | | 50 | 352 | 199 | 14 | 1 | 386 | | 2 | 2 |
| E470 | 18 | | 14 | 179 | 119 | 2 | | 70 | | | |
| E587 | 4 | 2 | 74 | | | 99 | 5 | | 5 | 21 | |
| F367 | 156 | | 131 | 239 | 381 | 31 | 1 | 18 | | 3 | |
| F726 | 17 | | 4 | 378 | 70 | 2 | | 474 | | 1 | 105 |
| F987 | 111 | | 82 | 662 | 302 | 52 | 8 | 270 | 13 | 24 | 60 |
| G045 | 10 | | 1 | 36 | 21 | | | 8 | | | |
| G012 | 16 | | 9 | 37 | 46 | 13 | | | | 2 | |
| G459 | | 2 | 12 | | | 96 | 18 | | 5 | 70 | |
| G794 | 8 | | 2 | 289 | 52 | 2 | 1 | 434 | | 1 | 22 |
| G854 | | 2 | | | | 17 | 35 | | 4 | 34 | |
| G545 | 62 | | 43 | 69 | 266 | 4 | | | | 5 | |
| G860 | 497 | 2 | 740 | 78 | 244 | 335 | 14 | | 4 | 39 | |
| H042 | 43 | | 18 | 167 | 187 | 4 | 2 | 94 | 1 | 3 | 12 |
| H110 | 113 | 1 | 108 | 958 | 347 | 53 | 9 | 606 | 4 | 27 | 98 |
| H262 | 10 | | 9 | 1 | 4 | 20 | 1 | | | 21 | |
| H263 | 18 | | 13 | 10 | 32 | 14 | | | 2 | 6 | |
| H497 | 5 | | | 337 | 46 | | | 218 | | | |
| H669 | 8 | | 22 | 361 | 31 | 1 | | 1149 | | | 193 |
| H670 | 22 | 6 | 23 | 18 | 29 | 33 | 18 | 4 | 16 | 38 | |
| H671 | 34 | | 39 | 319 | 112 | 14 | 2 | 151 | 2 | 6 | 3 |
| H672 | 18 | | 31 | 49 | 44 | 5 | 3 | 2 | | 4 | |
| H673 | 7 | | 28 | | 1 | 39 | 11 | | 6 | 22 | |
| H674 | 8 | 6 | 8 | 412 | 38 | 18 | 9 | 889 | 9 | 10 | 350 |
| H675 | 16 | | 20 | 12 | 20 | 77 | 1 | 6 | | 23 | 2 |
| H676 | 396 | 2 | 203 | 429 | 995 | 61 | 18 | 46 | 10 | 42 | |
| I442 | 89 | 1 | 64 | 801 | 180 | 70 | 2 | 1148 | 13 | 20 | 316 |
| L217 | | 27 | 3 | | | 15 | 75 | | 70 | 82 | |
| L582 | 18 | | 8 | 24 | 46 | 3 | | 2 | | 1 | 1 |
| L643 | 50 | | 46 | 107 | 98 | 11 | | 50 | | 3 | 1 |
| L647 | 23 | | 10 | 9 | 20 | 15 | 1 | 1 | | 9 | |
| L654 | | 38 | 37 | | | 217 | 201 | | 184 | 380 | |
| L783 | 98 | 4 | 125 | 39 | 120 | 158 | 47 | 4 | 21 | 90 | |
| C282 | 346 | 1 | 286 | 195 | 217 | 126 | 2 | 82 | 1 | 43 | 17 |
| L981 | 36 | | 20 | 246 | 100 | 5 | 1 | 281 | | 1 | 17 |
| VDA overall | 4172 | 237 | 5118 | 11,574 | 7744 | 4476 | 1141 | 15,594 | 1019 | 2490 | 10,210 |
| VDA overall % | 6.5 | 0.4 | 8.0 | 18.1 | 12.1 | 7.0 | 1.8 | 24.5 | 1.6 | 3.9 | 16.0 |
| ID Italian VDA Municipality | Cluster ID—Mean Gain LST Max—n° Population Exposed VDA Women | | | | | | | | | | |
| | I | II | III | IV | V | VI | VII | VIII | IX | X | XI |
| A205 | 17 | | 32 | | 10 | 41 | 3 | | 2 | 17 | |
| A305 | 2 | 7 | 40 | 1 | | 117 | 33 | | 19 | 110 | |
| A326 | 80 | 2 | 66 | 1423 | 273 | 31 | 3 | 6493 | 6 | 21 | 9365 |
| A424 | 172 | 2 | 233 | 9 | 50 | 82 | 35 | 1 | 29 | 61 | |
| A452 | 6 | | 5 | 261 | 88 | 2 | | 90 | | 1 | |
| A521 | 33 | | 16 | 49 | 73 | 4 | 11 | 1 | | 5 | |
| A094 | 15 | 27 | 49 | 1 | 2 | 79 | 148 | | 201 | 176 | |

**Table A2.** *Cont.*

| ID Italian VDA Municipality | Cluster ID—Mean Gain LST Max—n° Population Exposed VDA General | | | | | | | | | | |
|---|---|---|---|---|---|---|---|---|---|---|---|
| | **I** | **II** | **III** | **IV** | **V** | **VI** | **VII** | **VIII** | **IX** | **X** | **XI** |
| A108 | 16 | | 14 | 267 | 61 | 6 | | 606 | | 2 | 173 |
| A643 | 7 | | 12 | | | 15 | 9 | | | 15 | |
| A877 | 8 | | 26 | 11 | 15 | 20 | 4 | 3 | 10 | 8 | |
| B192 | 18 | | 26 | 209 | 85 | 17 | | 137 | | 1 | 4 |
| B230 | 4 | 3 | 74 | | | 161 | 29 | | 30 | 112 | |
| C593 | 94 | | 112 | 5 | 63 | 86 | 6 | | 2 | 32 | |
| C594 | 18 | | 6 | 148 | 70 | 4 | | 58 | | 1 | |
| C595 | 75 | 1 | 119 | 61 | 96 | 59 | 1 | 31 | | 9 | |
| B491 | | 1 | | | | 1 | 14 | | 15 | 17 | |
| C596 | 114 | 1 | 114 | 3 | 57 | 85 | 2 | | 1 | 9 | |
| B540 | | 31 | | | | 1 | 39 | | 61 | 53 | |
| C598 | 32 | | 33 | 306 | 83 | 15 | | 746 | | | 132 |
| C294 | 410 | 3 | 698 | 308 | 695 | 431 | 2 | 17 | 2 | 15 | |
| C821 | 121 | 5 | 140 | 20 | 170 | 126 | 32 | 2 | 16 | 74 | |
| D012 | 111 | | 77 | 526 | 477 | 36 | | 90 | | 1 | 64 |
| D338 | 36 | | 417 | | 1 | 674 | 15 | | 1 | 187 | |
| D356 | 20 | | 21 | 119 | 53 | 14 | 1 | 65 | | 4 | |
| D402 | 19 | 1 | 29 | | | 23 | 1 | | 2 | 39 | |
| D444 | 56 | 1 | 48 | 81 | 70 | 9 | | | | 2 | |
| D537 | 53 | | 14 | 486 | 307 | 8 | | 59 | | 1 | |
| D666 | 18 | 4 | 53 | | 2 | 95 | 11 | | 8 | 40 | |
| D839 | | 2 | 43 | | | 124 | 18 | | 3 | 64 | |
| E029 | 83 | 2 | 56 | 259 | 189 | 59 | 10 | 335 | 12 | 46 | 14 |
| E165 | 24 | | 29 | 114 | 77 | 34 | 1 | 864 | | 12 | 753 |
| E167 | 2 | 12 | 5 | | | 11 | 46 | | 73 | 22 | 1 |
| E168 | 1 | 11 | 8 | | | 73 | 102 | | 94 | 128 | |
| E273 | 25 | 5 | 154 | 12 | 14 | 257 | 30 | 8 | 11 | 96 | |
| E306 | 32 | | 26 | 114 | 157 | 11 | 1 | 5 | 1 | 4 | |
| E369 | 8 | 10 | 18 | | 1 | 78 | 16 | | 26 | 59 | |
| E371 | 177 | 13 | 169 | 5 | 208 | 80 | 21 | | 16 | 34 | |
| E391 | | | | 20 | | | | 247 | | | 171 |
| A308 | | 3 | | | | 4 | 10 | | 33 | 9 | |
| E458 | 82 | | 51 | 359 | 194 | 14 | 1 | 405 | | 2 | 3 |
| E470 | 17 | | 13 | 184 | 121 | 1 | | 72 | | | |
| E587 | 5 | 2 | 87 | | | 111 | 3 | | 5 | 20 | |
| F367 | 146 | | 120 | 232 | 355 | 29 | 1 | 17 | | 3 | |
| F726 | 15 | | 4 | 391 | 80 | 2 | | 502 | | 2 | 120 |
| F987 | 109 | | 77 | 706 | 311 | 51 | 7 | 289 | 9 | 22 | 60 |
| G045 | 12 | | 1 | 32 | 22 | | | 5 | | | |
| G012 | 14 | | 7 | 34 | 37 | 6 | | | | 1 | |
| G459 | | 1 | 14 | | | 107 | 13 | | 3 | 71 | |
| G794 | 7 | | 1 | 287 | 50 | 2 | 1 | 437 | | 1 | 22 |
| G854 | | 1 | | | | 20 | 32 | | 2 | 38 | |
| G545 | 64 | | 46 | 75 | 266 | 4 | | | | 4 | |
| G860 | 521 | 1 | 785 | 87 | 263 | 356 | 13 | | 3 | 32 | |
| H042 | 43 | | 20 | 151 | 178 | 4 | 2 | 87 | 1 | 3 | 13 |
| H110 | 107 | 1 | 100 | 954 | 345 | 49 | 8 | 602 | 2 | 24 | 84 |
| H262 | 8 | | 9 | | 3 | 13 | 1 | | | 11 | |
| H263 | 19 | | 14 | 10 | 33 | 14 | | | 1 | 5 | |
| H497 | 6 | | 1 | 333 | 46 | 1 | | 222 | | | |
| H669 | 9 | | 24 | 376 | 33 | 1 | | 1192 | | | 198 |
| H670 | 22 | 5 | 23 | 17 | 30 | 30 | 17 | 4 | 12 | 33 | |
| H671 | 35 | | 40 | 332 | 113 | 14 | 2 | 156 | 1 | 4 | 3 |
| H672 | 20 | | 24 | 49 | 50 | 5 | 2 | 2 | | 4 | |
| H673 | 7 | | 31 | | 1 | 43 | 13 | | 7 | 24 | |
| H674 | 9 | 5 | 9 | 418 | 38 | 16 | 8 | 883 | 8 | 11 | 350 |
| H675 | 13 | | 17 | 12 | 20 | 63 | 1 | 6 | | 19 | 2 |

**Table A2.** *Cont.*

| ID Italian VDA Municipality | Cluster ID—Mean Gain LST Max—n° Population Exposed VDA General | | | | | | | | | | |
|---|---|---|---|---|---|---|---|---|---|---|---|
| | I | II | III | IV | V | VI | VII | VIII | IX | X | XI |
| H676 | 405 | 2 | 200 | 456 | 1073 | 57 | 16 | 46 | 12 | 40 | |
| I442 | 89 | | 63 | 854 | 179 | 68 | 1 | 1190 | 3 | 22 | 323 |
| L217 | | 29 | 3 | | | 14 | 66 | | 65 | 77 | |
| L582 | 17 | | 7 | 24 | 41 | 1 | | 1 | | 1 | |
| L643 | 44 | | 41 | 95 | 86 | 11 | | 43 | | 3 | 1 |
| L647 | 28 | | 11 | 10 | 26 | 14 | | 1 | | 7 | |
| L654 | | 44 | 36 | | | 209 | 180 | | 174 | 337 | |
| L783 | 93 | 4 | 116 | 37 | 114 | 152 | 48 | 3 | 22 | 94 | |
| C282 | 368 | 1 | 305 | 199 | 232 | 139 | 1 | 85 | 1 | 48 | 18 |
| L981 | 34 | | 20 | 230 | 98 | 3 | | 271 | | | 17 |
| VDA overall | 4274 | 244 | 5304 | 11,763 | 7890 | 4597 | 1094 | 16,384 | 1008 | 2451 | 11,892 |
| VDA overall % | 6.4 | 0.4 | 7.9 | 17.6 | 11.8 | 6.9 | 1.6 | 24.5 | 1.5 | 3.7 | 17.8 |

| ID Italian VDA Municipality | Cluster ID—Mean Gain LST Max—n° Population Exposed VDA Women of Reproductive Age 15–49 | | | | | | | | | | |
|---|---|---|---|---|---|---|---|---|---|---|---|
| | I | II | III | IV | V | VI | VII | VIII | IX | X | XI |
| A205 | 8 | | 13 | | 7 | 12 | 1 | | | 4 | |
| A305 | 1 | 3 | 18 | | | 51 | 14 | | 9 | 51 | |
| A326 | 36 | 1 | 30 | 610 | 121 | 15 | 2 | 2504 | 3 | 10 | 3690 |
| A424 | 70 | 1 | 97 | 4 | 22 | 33 | 14 | 1 | 11 | 25 | |
| A452 | 3 | | 2 | 124 | 39 | 1 | | 45 | | | |
| A521 | 13 | | 6 | 19 | 29 | 1 | 5 | | | 2 | |
| A094 | 9 | 11 | 26 | 1 | 1 | 36 | 65 | | 76 | 76 | |
| A108 | 6 | | 6 | 121 | 26 | 4 | | 282 | | 1 | 81 |
| A643 | 3 | | 5 | | | 6 | 3 | | | 6 | |
| A877 | 4 | | 12 | 5 | 7 | 9 | 2 | 1 | 4 | 4 | |
| B192 | 8 | | 11 | 95 | 40 | 7 | | 62 | | | 1 |
| B230 | 2 | 1 | 32 | | | 66 | 12 | | 10 | 44 | |
| C593 | 43 | | 50 | 2 | 31 | 37 | 3 | | 1 | 14 | |
| C594 | 6 | | 3 | 74 | 28 | 2 | | 28 | | 1 | |
| C595 | 30 | | 48 | 26 | 39 | 25 | | 15 | | 4 | |
| B491 | | | | | | | 5 | | 5 | 7 | |
| C596 | 53 | | 50 | 2 | 30 | 37 | 1 | | 1 | 4 | |
| B540 | | 14 | | | | | 17 | | 29 | 22 | |
| C598 | 17 | | 18 | 143 | 42 | 9 | | 349 | | | 62 |
| C294 | 183 | 2 | 304 | 136 | 309 | 182 | 1 | 8 | 1 | 7 | |
| C821 | 48 | 2 | 58 | 7 | 69 | 53 | 12 | 1 | 6 | 30 | |
| D012 | 51 | | 35 | 238 | 214 | 16 | | 44 | | 1 | 33 |
| D338 | 16 | | 173 | | | 283 | 7 | | 1 | 81 | |
| D356 | 8 | | 10 | 46 | 20 | 7 | | 32 | | 1 | |
| D402 | 10 | 1 | 14 | | | 9 | 1 | | 1 | 17 | |
| D444 | 25 | | 20 | 38 | 32 | 2 | | | | 1 | |
| D537 | 23 | | 6 | 211 | 132 | 3 | | 26 | | | |
| D666 | 8 | 2 | 22 | | 1 | 36 | 4 | | 3 | 15 | |
| D839 | | 1 | 15 | | | 43 | 10 | | 1 | 24 | |
| E029 | 39 | 1 | 26 | 125 | 89 | 30 | 6 | 165 | 7 | 25 | 7 |
| E165 | 14 | | 17 | 57 | 41 | 20 | | 383 | | 7 | 343 |
| E167 | 1 | 6 | 2 | | | 5 | 20 | | 32 | 10 | |
| E168 | 1 | 4 | 4 | | | 33 | 44 | | 42 | 55 | |
| E273 | 11 | 2 | 76 | 5 | 6 | 118 | 13 | 4 | 5 | 41 | |
| E306 | 16 | | 12 | 49 | 71 | 5 | 1 | 2 | 1 | 2 | |
| E369 | 3 | 4 | 7 | | 1 | 31 | 6 | | 11 | 24 | |
| E371 | 67 | 5 | 65 | 2 | 79 | 35 | 11 | | 6 | 15 | |
| E391 | | | | 10 | | | | 129 | | | 86 |
| A308 | | 1 | | | | 2 | 5 | | 15 | 4 | |
| E458 | 32 | | 20 | 162 | 84 | 6 | | 173 | | 1 | 1 |
| E470 | 7 | | 6 | 76 | 53 | 1 | | 31 | | | |
| E587 | 2 | 1 | 34 | | | 47 | 2 | | 2 | 9 | |

**Table A2.** *Cont.*

| ID Italian VDA Municipality | Cluster ID—Mean Gain LST Max—n° Population Exposed VDA General | | | | | | | | | | |
|---|---|---|---|---|---|---|---|---|---|---|---|
| | I | II | III | IV | V | VI | VII | VIII | IX | X | XI |
| F367 | 69 | | 55 | 108 | 162 | 15 | 1 | 8 | | 2 | |
| F726 | 8 | | 2 | 175 | 36 | 1 | | 226 | | 1 | 54 |
| F987 | 48 | | 34 | 337 | 139 | 21 | 2 | 139 | 3 | 8 | 28 |
| G045 | 5 | | | 14 | 11 | | | 2 | | | |
| G012 | 7 | | 3 | 17 | 16 | 3 | | | | 1 | |
| G459 | | | 7 | | | 49 | 3 | | | 31 | |
| G794 | 3 | | 1 | 135 | 23 | 1 | | 205 | | | 10 |
| G854 | | 1 | | | | 7 | 12 | | 1 | 14 | |
| G545 | 31 | | 22 | 36 | 129 | 2 | | | | 1 | |
| G860 | 231 | 1 | 330 | 40 | 118 | 147 | 5 | | 1 | 14 | |
| H042 | 20 | | 9 | 73 | 82 | 2 | 1 | 45 | | 2 | 7 |
| H110 | 50 | | 44 | 458 | 164 | 21 | 3 | 291 | 1 | 11 | 38 |
| H262 | 4 | | 4 | | 1 | 6 | | | | 4 | |
| H263 | 6 | | 6 | 4 | 12 | 7 | | | 1 | 2 | |
| H497 | 3 | | | 161 | 23 | | | 104 | | | |
| H669 | 4 | | 11 | 173 | 16 | 1 | | 525 | | | 92 |
| H670 | 11 | 3 | 13 | 9 | 13 | 12 | 9 | 2 | 7 | 16 | |
| H671 | 15 | | 18 | 148 | 51 | 6 | 1 | 68 | 1 | 2 | 1 |
| H672 | 8 | | 14 | 22 | 22 | 2 | 1 | 1 | | 1 | |
| H673 | 3 | | 13 | | 1 | 18 | 5 | | 3 | 10 | |
| H674 | 5 | 2 | 5 | 195 | 20 | 7 | 4 | 420 | 4 | 6 | 166 |
| H675 | 6 | | 7 | 5 | 8 | 27 | | 2 | | 8 | |
| H676 | 171 | 1 | 88 | 195 | 431 | 26 | 7 | 22 | 4 | 18 | |
| I442 | 41 | | 26 | 411 | 83 | 29 | 1 | 567 | 1 | 11 | 147 |
| L217 | | 12 | 2 | | | 8 | 29 | | 28 | 34 | |
| L582 | 7 | | 3 | 7 | 14 | | | 1 | | | |
| L643 | 19 | | 18 | 46 | 40 | 5 | | 20 | | 1 | |
| L647 | 12 | | 4 | 4 | 11 | 7 | | | | 4 | |
| L654 | | 20 | 16 | | | 92 | 84 | | 78 | 156 | |
| L783 | 40 | 1 | 50 | 16 | 47 | 62 | 18 | 2 | 7 | 39 | |
| C282 | 147 | | 122 | 81 | 89 | 56 | | 38 | | 18 | 7 |
| L981 | 16 | | 10 | 102 | 45 | 2 | | 123 | | | 7 |
| VDA overall | 1866 | 106 | 2291 | 5360 | 3468 | 1958 | 475 | 7095 | 427 | 1062 | 4862 |
| VDA overall % | 6.4 | 0.4 | 7.9 | 18.5 | 12.0 | 6.8 | 1.6 | 24.5 | 1.5 | 3.7 | 16.8 |

| ID Italian VDA Municipality | Cluster ID—Mean Gain LST Max—n° Population Exposed VDA Elderly 60 Plus | | | | | | | | | | |
|---|---|---|---|---|---|---|---|---|---|---|---|
| | I | II | III | IV | V | VI | VII | VIII | IX | X | XI |
| A205 | 10 | | 19 | | 1 | 35 | 3 | | 1 | 13 | |
| A305 | 1 | 3 | 21 | | | 61 | 18 | | 10 | 55 | |
| A326 | 39 | 1 | 31 | 775 | 143 | 14 | 1 | 4048 | 2 | 9 | 5644 |
| A424 | 102 | 1 | 131 | 5 | 26 | 44 | 17 | 1 | 14 | 30 | |
| A452 | 5 | | 5 | 125 | 48 | 2 | | 41 | | 1 | |
| A521 | 16 | | 8 | 19 | 33 | 3 | 3 | 1 | | 2 | |
| A094 | 8 | 18 | 27 | | 1 | 38 | 73 | | 106 | 88 | |
| A108 | 9 | | 6 | 130 | 34 | 2 | | 306 | | 1 | 88 |
| A643 | 5 | | 9 | | | 11 | 6 | | | 11 | |
| A877 | 4 | | 14 | 6 | 8 | 11 | 3 | 1 | 6 | 4 | |
| B192 | 8 | | 14 | 96 | 38 | 11 | | 62 | | 1 | 2 |
| B230 | 2 | 2 | 37 | | | 84 | 17 | | 20 | 65 | |
| C593 | 51 | | 69 | 4 | 32 | 54 | 4 | | 2 | 24 | |
| C594 | 13 | | 3 | 76 | 47 | 3 | | 34 | | 1 | |
| C595 | 46 | 1 | 71 | 35 | 57 | 36 | 1 | 16 | | 6 | |
| B491 | | 1 | | | | 1 | 10 | | 11 | 11 | |
| C596 | 48 | | 55 | 1 | 23 | 44 | 1 | | | 5 | |
| B540 | | 19 | | | | 1 | 28 | | 40 | 38 | |
| C598 | 13 | | 12 | 138 | 35 | 5 | | 333 | | | 58 |
| C294 | 223 | 1 | 376 | 168 | 381 | 247 | | 6 | | 5 | |

**Table A2.** *Cont.*

| ID Italian VDA Municipality | \| Cluster ID—Mean Gain LST Max—n° Population Exposed VDA General |||||||||| |
|---|---|---|---|---|---|---|---|---|---|---|---|
| | I | II | III | IV | V | VI | VII | VIII | IX | X | XI |
| C821 | 80 | 3 | 90 | 15 | 119 | 82 | 21 | 1 | 11 | 49 | |
| D012 | 57 | | 43 | 263 | 254 | 21 | | 37 | | 1 | 27 |
| D338 | 18 | | 231 | | | 372 | 9 | | 1 | 104 | |
| D356 | 11 | | 12 | 71 | 30 | 8 | 1 | 31 | | 5 | |
| D402 | 9 | | 16 | | | 16 | 1 | | 1 | 30 | |
| D444 | 28 | | 25 | 43 | 39 | 7 | | | | 2 | |
| D537 | 29 | | 8 | 251 | 161 | 4 | | 30 | | | |
| D666 | 13 | 3 | 36 | | 1 | 63 | 8 | | 5 | 28 | |
| D839 | | 3 | 29 | | | 86 | 9 | | 6 | 42 | |
| E029 | 35 | | 23 | 102 | 78 | 26 | 3 | 133 | 3 | 17 | 6 |
| E165 | 9 | | 12 | 53 | 30 | 14 | | 463 | | 5 | 354 |
| E167 | 1 | 4 | 3 | | | 6 | 20 | | 28 | 8 | |
| E168 | 1 | 4 | 4 | | | 40 | 59 | | 47 | 75 | |
| E273 | 14 | 2 | 80 | 7 | 8 | 143 | 17 | 5 | 6 | 55 | |
| E306 | 15 | | 12 | 59 | 75 | 5 | | 3 | | 2 | |
| E369 | 3 | 6 | 8 | | 1 | 45 | 10 | | 16 | 38 | |
| E371 | 104 | 8 | 97 | 3 | 128 | 44 | 10 | | 9 | 19 | |
| E391 | | | | 8 | | | | 101 | | | 71 |
| A308 | | 1 | | | | 3 | 7 | | 18 | 6 | |
| E458 | 42 | | 24 | 172 | 104 | 8 | | 196 | | 1 | 2 |
| E470 | 8 | | 7 | 89 | 56 | 1 | | 38 | | | |
| E587 | 3 | 1 | 54 | | | 67 | 3 | | 3 | 13 | |
| F367 | 71 | | 60 | 105 | 185 | 13 | | 8 | | 1 | |
| F726 | 4 | | 1 | 200 | 37 | 1 | | 242 | | | 55 |
| F987 | 53 | | 39 | 292 | 137 | 29 | 4 | 115 | 7 | 12 | 26 |
| G045 | 7 | | 1 | 20 | 14 | | | 3 | | | |
| G012 | 7 | | 4 | 13 | 17 | 5 | | | | 1 | |
| G459 | | 2 | 7 | | | 56 | 14 | | 4 | 43 | |
| G794 | 4 | | 1 | 133 | 25 | 1 | | 193 | | 1 | 10 |
| G854 | | | | | | 14 | 25 | | 1 | 25 | |
| G545 | 29 | | 21 | 34 | 122 | 2 | | | | 2 | |
| G860 | 288 | | 440 | 48 | 147 | 198 | 7 | | 2 | 21 | |
| H042 | 23 | | 11 | 79 | 98 | 1 | | 43 | 1 | 1 | 6 |
| H110 | 49 | 1 | 49 | 431 | 158 | 26 | 5 | 273 | 4 | 14 | 43 |
| H262 | 3 | | 3 | | 1 | 8 | 1 | | | 9 | |
| H263 | 15 | | 6 | 7 | 24 | 5 | | | 1 | 2 | |
| H497 | 2 | | | 149 | 19 | | | 100 | | | |
| H669 | 4 | | 10 | 186 | 16 | 1 | | 610 | | | 102 |
| H670 | 12 | 3 | 9 | 7 | 15 | 18 | 9 | 2 | 5 | 19 | |
| H671 | 20 | | 22 | 171 | 58 | 10 | | 85 | | 3 | 2 |
| H672 | 10 | | 10 | 26 | 24 | 3 | 2 | 1 | | 2 | |
| H673 | 4 | | 17 | | 1 | 24 | 7 | | 4 | 13 | |
| H674 | 4 | 3 | 3 | 206 | 20 | 11 | 4 | 418 | 4 | 4 | 167 |
| H675 | 8 | | 11 | 6 | 12 | 41 | | 4 | | 12 | 1 |
| H676 | 234 | | 114 | 265 | 646 | 31 | 5 | 24 | 4 | 20 | |
| I442 | 44 | | 34 | 389 | 87 | 35 | 1 | 565 | 5 | 8 | 165 |
| L217 | | 17 | 2 | | | 9 | 41 | | 41 | 50 | |
| L582 | 11 | | 5 | 16 | 29 | 1 | | 1 | | 1 | |
| L643 | 25 | | 24 | 49 | 47 | 5 | | 24 | | 1 | |
| L647 | 14 | | 9 | 6 | 14 | 9 | | 1 | | 3 | |
| L654 | | 20 | 20 | | | 119 | 87 | | 87 | 160 | |
| L783 | 51 | 2 | 68 | 22 | 64 | 95 | 31 | 2 | 15 | 59 | |
| C282 | 204 | 1 | 168 | 123 | 150 | 75 | 2 | 44 | 1 | 26 | 11 |
| L981 | 18 | | 8 | 116 | 52 | 2 | | 130 | | | 9 |

**Table A2.** *Cont.*

| ID Italian VDA Municipality | Cluster ID—Mean Gain LST Max—n° Population Exposed VDA General | | | | | | | | | | |
|---|---|---|---|---|---|---|---|---|---|---|---|
| | I | II | III | IV | V | VI | VII | VIII | IX | X | XI |
| VDA overall | 2299 | 135 | 2901 | 5816 | 4210 | 2613 | 611 | 8772 | 553 | 1383 | 6851 |
| VDA overall % | 6.4 | 0.4 | 8.0 | 16.1 | 11.6 | 7.2 | 1.7 | 24.3 | 1.5 | 3.8 | 19.0 |
| ID Italian VDA Municipality | Cluster ID—Mean Gain LST Max—n° Population Exposed Children under 5 | | | | | | | | | | |
| | I | II | III | IV | V | VI | VII | VIII | IX | X | XI |
| A205 | 2 | | 6 | | 1 | 4 | | | | 3 | |
| A305 | | 2 | 3 | | | 12 | 5 | | 2 | 14 | |
| A326 | 9 | | 10 | 127 | 27 | 3 | | 520 | 1 | 2 | 732 |
| A424 | 15 | | 20 | 1 | 4 | 6 | 3 | | 2 | 6 | |
| A452 | | | | 29 | 8 | | | 11 | | | |
| A521 | 5 | | 3 | 10 | 13 | 1 | 3 | | | 2 | |
| A094 | 2 | 5 | 8 | | | 12 | 15 | | 29 | 21 | |
| A108 | 1 | | | 27 | 5 | | | 57 | | | 16 |
| A643 | | | 1 | | | 1 | 1 | | | 1 | |
| A877 | 1 | | 3 | 1 | 2 | 2 | 1 | | 1 | 1 | |
| B192 | 1 | | 2 | 19 | 8 | | | 13 | | | |
| B230 | | | 10 | | | 15 | 2 | | 3 | 9 | |
| C593 | 8 | | 9 | | 8 | 9 | | | | 4 | |
| C594 | 1 | | | 15 | 5 | 1 | | 6 | | | |
| C595 | 7 | | 9 | 5 | 9 | 4 | | 2 | | 1 | |
| B491 | | | | | | | 1 | | 1 | 1 | |
| C596 | 9 | | 9 | | | 6 | 7 | | | 1 | |
| B540 | | 1 | | | | | 2 | | 5 | 3 | |
| C598 | 4 | | 4 | 34 | 10 | 2 | | 82 | | | 14 |
| C294 | 39 | 1 | 70 | 33 | 76 | 32 | | 2 | | 1 | |
| C821 | 9 | | 11 | 1 | 12 | 9 | 2 | | 1 | 4 | |
| D012 | 9 | | 5 | 50 | 44 | 2 | | 8 | | | 8 |
| D338 | 3 | | 34 | | | 66 | 1 | | | 19 | |
| D356 | 2 | | 2 | 10 | 4 | 2 | | 8 | | | |
| D402 | 1 | | 3 | | | 4 | | | | 6 | |
| D444 | 7 | | 7 | 6 | 6 | 1 | | | | | |
| D537 | 5 | | 2 | 56 | 30 | 1 | | 8 | | | |
| D666 | 1 | | 4 | | | 9 | 1 | | 1 | 4 | |
| D839 | | | 2 | | | 8 | 3 | | | 4 | |
| E029 | 8 | | 7 | 31 | 21 | 8 | 1 | 45 | 1 | 5 | 2 |
| E165 | 2 | | 2 | 11 | 7 | 3 | | 87 | | 1 | 91 |
| E167 | | 2 | 1 | | | 1 | 7 | | 10 | 4 | |
| E168 | | 2 | 1 | | | 8 | 7 | | 9 | 10 | |
| E273 | 2 | 1 | 7 | 1 | 1 | 17 | 3 | 1 | 1 | 8 | |
| E306 | 4 | | 4 | 16 | 20 | 1 | | 1 | | 1 | |
| E369 | 1 | 1 | 2 | | | 10 | 2 | | 4 | 6 | |
| E371 | 17 | 2 | 17 | | 19 | 8 | 3 | | 2 | 4 | |
| E391 | | | | 2 | | | | 28 | | | 20 |
| A308 | | | | | | 1 | 1 | | 3 | 1 | |
| E458 | 11 | | 7 | 41 | 23 | 1 | | 49 | | | |
| E470 | 1 | | 1 | 16 | 10 | | | 7 | | | |
| E587 | | | 5 | | | 7 | 1 | | 1 | 2 | |
| F367 | 12 | | 12 | 23 | 35 | 3 | | 2 | | | |
| F726 | 3 | | 1 | 39 | 9 | | | 48 | | | 10 |
| F987 | 12 | | 8 | 76 | 31 | 3 | 1 | 31 | 1 | 1 | 7 |
| G045 | 1 | | | 2 | 2 | | | | | | |
| G012 | 2 | | 1 | 5 | 5 | | | | | 1 | |
| G459 | | | 1 | | | 8 | 1 | | | 7 | |
| G794 | 1 | | | 27 | 4 | | | 48 | | | 2 |
| G854 | | | | | | 3 | | | | 4 | |
| G545 | 4 | | 3 | 4 | 20 | | | | | | |
| G860 | 50 | | 69 | 8 | 24 | 31 | 1 | | | 1 | |

| ID Italian VDA Municipality | Cluster ID—Mean Gain LST Max—n° Population Exposed VDA General | | | | | | | | | | |
|---|---|---|---|---|---|---|---|---|---|---|---|
| | I | II | III | IV | V | VI | VII | VIII | IX | X | XI |
| H042 | 5 | | 2 | 16 | 18 | 1 | | 9 | | 1 | 1 |
| H110 | 12 | | 10 | 114 | 40 | 5 | 1 | 70 | | 2 | 9 |
| H262 | 1 | | 1 | | | 3 | | | | 3 | |
| H263 | 1 | | 2 | 1 | 2 | 2 | | | | 1 | |
| H497 | 1 | | | 35 | 6 | | | 21 | | | |
| H669 | 1 | | 3 | 45 | 4 | | | 122 | | | 18 |
| H670 | 2 | | 4 | 2 | 3 | 3 | 2 | | 1 | 4 | |
| H671 | 3 | | 6 | 31 | 13 | 1 | 1 | 15 | | 1 | |
| H672 | 3 | | 3 | 4 | 4 | 1 | | | | | |
| H673 | 1 | | 2 | | | 3 | 1 | | | 2 | |
| H674 | 1 | 1 | 2 | 42 | 5 | 1 | 2 | 95 | 2 | 2 | 45 |
| H675 | 1 | | 1 | 2 | 2 | 4 | | 1 | | 1 | |
| H676 | 30 | | 17 | 29 | 76 | 4 | 3 | 4 | | 4 | |
| I442 | 10 | | 8 | 85 | 19 | 8 | | 114 | | 3 | 33 |
| L217 | | 3 | | | | 1 | 7 | | 6 | 7 | |
| L582 | | | | 1 | 2 | | | | | | |
| L643 | 6 | | 5 | 11 | 11 | 1 | | 5 | | | |
| L647 | 2 | | 1 | | 2 | 2 | | | | 2 | |
| L654 | | 4 | 4 | | | 21 | 19 | | 19 | 33 | |
| L783 | 15 | 1 | 17 | 4 | 18 | 17 | 4 | | 3 | 8 | |
| C282 | 39 | | 31 | 16 | 18 | 13 | | 7 | | 5 | 1 |
| L981 | 2 | | 1 | 21 | 9 | | | 33 | | | 3 |
| VDA overall | 409 | 30 | 504 | 1190 | 760 | 420 | 110 | 1562 | 114 | 241 | 1013 |
| VDA overall % | 6.4 | 0.5 | 7.9 | 18.7 | 12.0 | 6.6 | 1.7 | 24.6 | 1.8 | 3.8 | 15.9 |

| ID Italian VDA Municipality | Cluster ID—Mean Gain LST Max—n° Population Exposed VDA Youth 15–24 | | | | | | | | | | |
|---|---|---|---|---|---|---|---|---|---|---|---|
| | I | II | III | IV | V | VI | VII | VIII | IX | X | XI |
| A205 | 3 | | 3 | | 2 | 3 | | | | 1 | |
| A305 | | 2 | 5 | | | 20 | 5 | | 3 | 16 | |
| A326 | 17 | | 15 | 262 | 56 | 7 | 1 | 1044 | 2 | 6 | 1493 |
| A424 | 26 | | 39 | 2 | 10 | 16 | 6 | | 5 | 11 | |
| A452 | 1 | | | 53 | 19 | | | 17 | | | |
| A521 | 4 | | 2 | 9 | 9 | | 1 | | | | |
| A094 | 4 | 3 | 10 | | | 12 | 34 | | 32 | 34 | |
| A108 | 2 | | 1 | 39 | 8 | | | 97 | | | 28 |
| A643 | 1 | | 2 | | | 2 | 1 | | | 2 | |
| A877 | 1 | | 6 | 1 | 2 | 4 | | | | 1 | |
| B192 | 3 | | 5 | 37 | 15 | 5 | | 24 | | | 1 |
| B230 | 1 | 1 | 14 | | | 26 | 6 | | 6 | 18 | |
| C593 | 23 | | 23 | 1 | 15 | 17 | 1 | | | 4 | |
| C594 | 2 | | 1 | 27 | 10 | | | 10 | | | |
| C595 | 13 | | 19 | 15 | 20 | 8 | | 9 | | 1 | |
| B491 | | | | | | | 2 | | 2 | 3 | |
| C596 | 19 | | 20 | 1 | 9 | 14 | | | | 1 | |
| B540 | | 6 | | | | | 9 | | 13 | 14 | |
| C598 | 7 | | 7 | 57 | 18 | 4 | | 136 | | | 29 |
| C294 | 76 | 1 | 126 | 58 | 127 | 82 | | 4 | | 2 | |
| C821 | 18 | 1 | 21 | 3 | 26 | 19 | 5 | | 2 | 9 | |
| D012 | 23 | | 16 | 100 | 88 | 6 | | 19 | | | 14 |
| D338 | 7 | | 65 | | | 112 | 3 | | | 35 | |
| D356 | 3 | | 5 | 19 | 9 | 3 | | 12 | | | |
| D402 | 1 | | 3 | | | 2 | | | | 7 | |
| D444 | 9 | | 8 | 14 | 13 | 2 | | | | | |
| D537 | 8 | | 2 | 69 | 46 | 1 | | 10 | | | |
| D666 | 3 | 1 | 10 | | | 12 | 2 | | 2 | 6 | |
| D839 | | | 6 | | | 15 | 3 | | | 10 | |
| E029 | 17 | | 11 | 42 | 34 | 11 | 2 | 53 | 2 | 8 | 2 |

**Table A2.** *Cont.*

| ID Italian VDA Municipality | I | II | III | IV | V | VI | VII | VIII | IX | X | XI |
|---|---|---|---|---|---|---|---|---|---|---|---|
| E165 | 4 | | 6 | 19 | 12 | 6 | | 140 | | 2 | 119 |
| E167 | 1 | 2 | 2 | | | 4 | 7 | | 9 | 6 | |
| E168 | | 2 | 1 | | | 12 | 19 | | 20 | 22 | |
| E273 | 4 | 1 | 31 | 2 | 3 | 46 | 5 | 2 | 2 | 16 | |
| E306 | 8 | | 5 | 21 | 35 | 3 | | | | 1 | |
| E369 | 1 | 1 | 2 | | | 13 | 2 | | 3 | 11 | |
| E371 | 27 | 1 | 24 | 1 | 34 | 15 | 4 | | 2 | 5 | |
| E391 | | | | 4 | | | | 54 | | | 34 |
| A308 | | | | | | 1 | 2 | | 6 | 2 | |
| E458 | 12 | | 7 | 57 | 29 | 2 | | 67 | | | 1 |
| E470 | 4 | | 3 | 37 | 28 | | | 15 | | | |
| E587 | 1 | | 15 | | | 18 | 1 | | | 3 | |
| F367 | 28 | | 22 | 41 | 60 | 5 | | 3 | | | |
| F726 | 2 | | 1 | 67 | 13 | | | 86 | | | 22 |
| F987 | 20 | | 15 | 132 | 57 | 8 | | 56 | | 3 | 11 |
| G045 | 2 | | | 2 | 3 | | | | | | |
| G012 | 1 | | | 6 | 8 | | | | | | |
| G459 | | | 2 | | | 16 | 1 | | | 13 | |
| G794 | 2 | | | 63 | 11 | | | 89 | | | 5 |
| G854 | | | | | | 2 | 10 | | 1 | 6 | |
| G545 | 14 | | 11 | 18 | 55 | 1 | | | | 1 | |
| G860 | 93 | | 141 | 15 | 47 | 60 | 3 | | 1 | 7 | |
| H042 | 7 | | 3 | 31 | 34 | 1 | | 17 | | | 2 |
| H110 | 20 | | 17 | 168 | 65 | 6 | 1 | 112 | | 3 | 15 |
| H262 | 2 | | 2 | | 1 | 3 | | | | 2 | |
| H263 | 2 | | 4 | 2 | 4 | 4 | | | | 1 | |
| H497 | 1 | | | 70 | 9 | | | 45 | | | |
| H669 | 2 | | 5 | 63 | 7 | | | 177 | | | 26 |
| H670 | 5 | 1 | 4 | 4 | 7 | 4 | 1 | 1 | 1 | 3 | |
| H671 | 8 | | 9 | 56 | 21 | 2 | | 25 | | 1 | |
| H672 | 2 | | 8 | 8 | 8 | 1 | 1 | | | | |
| H673 | 1 | | 6 | | | 9 | 3 | | 1 | 5 | |
| H674 | 2 | 1 | 2 | 77 | 7 | 4 | 2 | 155 | 2 | 2 | 52 |
| H675 | 2 | | 3 | 1 | 3 | 12 | | | | 4 | |
| H676 | 69 | | 33 | 80 | 172 | 10 | 2 | 10 | | 4 | |
| I442 | 18 | | 10 | 156 | 36 | 10 | 1 | 230 | | 3 | 59 |
| L217 | | 3 | 2 | | | 4 | 10 | | 11 | 13 | |
| L582 | 3 | | 1 | 5 | 6 | | | 1 | | | |
| L643 | 10 | | 9 | 20 | 19 | 3 | | 9 | | 1 | |
| L647 | 3 | | 1 | 1 | 4 | 1 | | | | 1 | |
| L654 | | 8 | 6 | | | 34 | 34 | | 35 | 62 | |
| L783 | 10 | | 13 | 4 | 13 | 17 | 5 | 1 | 2 | 14 | |
| C282 | 61 | | 50 | 37 | 42 | 20 | | 17 | | 7 | 3 |
| L981 | 9 | | 6 | 48 | 21 | 1 | | 51 | | | 2 |
| VDA overall | 753 | 39 | 926 | 2122 | 1408 | 767 | 196 | 2799 | 168 | 417 | 1918 |
| VDA overall % | 6.5 | 0.3 | 8.0 | 18.4 | 12.2 | 6.7 | 1.7 | 24.3 | 1.5 | 3.6 | 16.7 |

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
