# Peer review of "Risk Assessment of Rising Temperatures Using Landsat 4–9 LST Time Series and Meta® Population Dataset: An Application in Aosta Valley, NW Italy"

_remotesensing, doi:10.3390/rs15092348_

Round 1

Reviewer 1 Report

Dear Authors,

English improvement is required, and lengthy tables should be moved to the appendix. Please revise and adjust your manuscript based on my comments, which are written on the uploaded pdf file to the journal.

Good luck.

Dear Authors,

English improvement is required, and lengthy tables should be moved to the appendix. Please revise and adjust your manuscript based on my comments, which are written on the uploaded pdf file to the journal.

Good luck.

Author Response

Response to Reviewer 1 Comments

We would like to thank reviewers for their appropriate comments and helpful suggestions that have been carefully considered. Majority of provided suggestions highlighted gaps in the text and were really useful to improve, we hope, paper quality. Referees can find their comment and authors’ actions to reply/satisfy requests.

In particular, the synthesis of reviewers’ comments suggested a deep revision in paper organization and harmonization. Consequently, you will find some structural changes aimed at simplifying paper reading and make content more effective.

All comments were carefully evaluated and for the most of them corrections and integrations have been provided. Thank you so much for your work!

Point 1: English improvement is required, and lengthy tables should be moved to the appendix. Please revise and adjust your manuscript based on my comments, which are written on the uploaded pdf file to the journal.

Response 1: Firstly, we would like to thanks the reviewer for his/her suggestion and comments. A general comprehensive English editing has been done taking into account also the suggestion proposed by the other reviewers therefore all the several points proposed have been changed.  Please see into the manuscript the changed performed. Finally as suggested many tables have been moved into appendix or supplementary materials.

Reviewer 2 Report

The study made an assessment on Exposure-risk by using LST derived from Landsat data along with the population dataset. As per the current form, the manuscript needs improvement. The specific comments are given below.

 Major Comments:

1)      Abstract: It is not written concisely. Many of the unimportant aspects are kept here (for instance, in L25 related to SAGA GIS … )

2)      Introduction is not properly contextualized. It needs great improvement especially related to studies on risk assessment  

3)      Objective of the study is not clearly defined like why solar radiation is not included here

4)      Novelty of the study is not clearly defined before writing the objectives

5)      Fig 1 is quite similar to Fig 3 & so Fig 1 can be replaced by providing some important location information or land use land coper map.

6)      Results: Table 5 shall be moved to an appendix. Also, it is not clear why so many the gap in Table 5  

7)      Figure 6: zoomed regions do not have latitude and longitude information though it was present in Figure 4. Moreover, in figure 3, all zoomed regions used in Figs 4 & 6 need to be marked in Figure 3 by rectangular boxes. It will help readers to locate it.   

8)       Discussion needs improvement by including validation & limitation of the study.

Minor comments

·         L84: "the native geometric resolution for the thermal bands of the Landsat missions". This has to be moved from here as it is not related to population para.

·         Many abbreviations are used but never expanded when it was used for first time 

Writing needs some revision

Author Response

Response to Reviewer 2 Comments

We would like to thank reviewers for their appropriate comments and helpful suggestions that have been carefully considered. Majority of provided suggestions highlighted gaps in the text and were really useful to improve, we hope, paper quality. Referees can find their comment and authors’ actions to reply/satisfy requests.

In particular, the synthesis of reviewers’ comments suggested a deep revision in paper organization and harmonization. Consequently, you will find some structural changes aimed at simplifying paper reading and make content more effective.

All comments were carefully evaluated and for the most of them corrections and integrations have been provided. Thank you so much for your work!

Point 1: Summary:

The study made an assessment on Exposure-risk by using LST derived from Landsat data along with the population dataset. As per the current form, the manuscript needs improvement. The specific comments are given below.

 Major Comments:

1)      Abstract: It is not written concisely. Many of the unimportant aspects are kept here (for instance, in L25 related to SAGA GIS … )

2)      Introduction is not properly contextualized. It needs great improvement especially related to studies on risk assessment 

3)      Objective of the study is not clearly defined like why solar radiation is not included here

4)      Novelty of the study is not clearly defined before writing the objectives

5)      Fig 1 is quite similar to Fig 3 & so Fig 1 can be replaced by providing some important location information or land use land coper map.

6)      Results: Table 5 shall be moved to an appendix. Also, it is not clear why so many the gap in Table 5 

7)      Figure 6: zoomed regions do not have latitude and longitude information though it was present in Figure 4. Moreover, in figure 3, all zoomed regions used in Figs 4 & 6 need to be marked in Figure 3 by rectangular boxes. It will help readers to locate it.  

8)       Discussion needs improvement by including validation & limitation of the study.

Minor comments

  • L84: "the native geometric resolution for the thermal bands of the Landsat missions". This has to be moved from here as it is not related to population para.

  • Many abbreviations are used but never expanded when it was used for first time

Response 1: Firstly, we would like to thank the reviewer for his/her valuable comments. We have followed the suggestion and removed some parts as reported:

  • The abstarct has been changed in order to be much clear and concise as follow: “Earth Observation Data has assumed a key role in environmental monitoring, as well as, in risk assessment. Rising temperatures and consequently heat waves due to ongoing climate change represent an important risk considering the population, as well as animals exposed. Therefore, this study was focused on the Aosta Valley Region in NW Italy. To assess population exposure to these patterns, the following datasets have been considered (1) HDX Meta population dataset re-fined and updated in order to map population distribution and their features; (2) Landsat collec-tion (missions 4 to 9) from 1984 to 2022 obtained and calibrated in Google Earth Engine to model LST trends. A pixel-based analysis was performed considering Aosta Valley settlements and rel-ative population distribution according to the Meta population dataset. From Landsat data, LST trends were modelled. The LST gains computed were used to produce risk exposure maps con-sidering the population distribution and structure (like ages, gender etc). To check the con-sistency and quality of HDX population dataset MAE was computed considering the ISTAT pop-ulation dataset at the municipality level. Exposure-risk maps were finally realized adopting two different approaches. The first one considers only LST gain maximum by performing an ISO-DATA unsupervised classification clustering in which the separability of each class obtained was checked by computing the Jeffries-Matusita (J-M) distances. The second one, by developing a suitability risk map modeler to map the rising temperature exposure. In this last case the input parameters considered were defined after performing a multivariate regression in which LST maximum was correlated and tested considering a) Fractional Vegetation Cover (FVC), b) Quote, c) Slope, d) Aspect e) Potential Incoming Solar Radiation (mean sun light duration in the meteorological summer season) f) LST gain mean. Results show a steeper increase in LST maxi-mum trend, especially in the bottom valley municipalities. Especially in new built-up areas around factories where more than 60% of the Aosta Valley population and domestic animals live and where a high exposure has been detected and mapped with both approaches performed. Maps produced may help the local planners and the civil protection to face global warming in a One Health perspective.”

  • We are partially agreed with the reviewer we have contextalized the EO data and problems with relatives maps of population but as suggested by the referee, we have strongly improved in the introduction the studies on risk assessment and divided the introduction in subchapter please see into the introduction section into the manuscript. Here we report the risk assessment part:

“1.3 Remote sensing in climate change risk assessment

There is a growing need for the assessment and reduction of climate change risk. The effects of global warming are already bringing harm to human communities and the nat-ural world. Further temperature rises will have a devastating impact and more action on greenhouse gas emissions is urgently required. Multiple factors contribute to climate change, and multiple actions are needed to address it [8,37–41]. Especially concerning on population exposure to climate change. In fact, nowadays, EO Data and more generally remote sensing may help in mapping and developing services with particular regard onto climate change risk assessment involving communities at different level. Space-borne images for civil applications have been routinely acquired since the 1980s (Landsat and SPOT), while more recently, the European Union’s Copernicus program. EO data can pro-vide remotely sensed information regarding floods, forest fires, and droughts. In general, remote sensing data from space, but also from airborne or drone platforms, can be profitably used to manage many risks, from geo-hydrological to volcanic, from seismic to anthropogenic. Less exploited is the application and coupling of remote sensing data with GIS health data with particular regard onto the climate change framework. Remote sensing can play a key role in managing risks, leading to a new level of understanding of the complex Earth systems and planning. In recent decades, satellite-based observations and the derived geospatial products have been successfully demonstrated to be highly valuable tools in each different phase of the risk and exposure management (forecasting, planning, emergency, and post-emergency) [34,42]. For example, synthetic aperture radar (SAR) can facilitate risk management since they are also acquired through dense cloud cover and in both night and day conditions. This ability can help during the emergency phase. Stacks of SAR data can be used to detect subtle ground deformation induced by slow movement phenomena (e.g., slow landslides, subsidence) that can dangerously evolve, involving elements of risk [62]. On the other hand, optical images are fundamental products to monitor land cover changes induced by several hazards (e.g., fast landslides, volcanic eruptions) or thermal data to assess for eg. Urban heath island (UHI) and their intensity or the water stress onto forests and crops. These data are routinely used to map and evaluate the element at risk scattered over wide areas. The application of a combined used of population data at higher resolution with thermal EO Data in order to evaluate the exposure to rising temperature in light of climate change has poorly explored in the scientific community. This is due to the fact that the population dataset at higher resolution are relatively new and also the application of EO Data in the domain. of the climate change adaptation regarding the civil component are moving their first steps.

Moreover, the One Health approach involving thermal remote sensed time-series analysis to assess the temperature trend gain represent a novelty than the well-know and over-studied UHI phenomenon which is focused only on a given time and do not permit to develop strong models. The LST trends analysis modelling and their application coupling population data represent a novelty especially in the assessment of rising temperature exposure [8].  

  • The referee is right we have better explain into the text as follow: “5 Aims

Finally, the main aim of this work was to perform a risk population assessment to rising temperatures and heat-waves by Landsat Land Surface Temperature timeseries in Aosta Valley, NW Italy by realizing a scalable application to each 18 countries that al-ready have HDX Meta dataset. The analysis was performed at a pixel level grouping the final population exposure at a municipality level. It is worth noting that the map generat-ed will be available at a pixel-level. Moreover, the quality of the population dataset was checked and a risk map performed considering the population distribution and the LST gains modelized. In particular, LST maximum and mean trends were computed consid-ering their significance and possible correlations were tested considering the following parameters: a) Fractional Vegetation Cover (FVC), b) Quote, c) Slope, d) Aspect e) Potential Incoming Solar Radiation (mean sun light duration in the meteorological summer season) in order to assess which parameters include into the risk model. The final outputs have permitted to map and assess the LST gain in the last 39 years (1984-2022) and relative population exposure to LST trends per ages bands and gender groups providing hopefully useful information to civil protection and the health sector permitting them to detect areas in which call to health emergencies would be more likely during heatwaves and urban planners to promote greening actions in a mitigation and adaptation perspective to climate change according a One Health approach.”

  • The referee is right, in order to be much clear, we have rearranged the abstract and the introduction to quickly point out the objectives and aims as follow:

“The application of a combined used of population data at higher resolution with thermal EO Data in order to evaluate the exposure to rising temperature in light of climate change has poorly explored in the scientific community. This is due to the fact that the population dataset at higher resolution are relatively new and also the application of EO Data in the domain. of the climate change adaptation regarding the civil component are moving their first steps.

Moreover, the One Health approach involving thermal remote sensed time-series analysis to assess the temperature trend gain represent a novelty than the well-know and over-studied UHI phenomenon which is focused only on a given time and do not permit to develop strong models. The LST trends analysis modelling and their application cou-pling population data represent a novelty especially in the assessment of rising tempera-ture exposure [8].  

1.4 Coupling population and EO Data in climate change adaptation and risk assessment

The approach developed to map population (thanks to Meta Geo For Good) coupling free thermal EO Data to model rising temperature in order to map the exposure and risk considering population age bands and gender groups with all dataset involved with the same native geometrical resolution (GSD 30 m) enforces the applicability and coupling of these kind of data in the planning and management of climate change risks and adapta-tion suggesting new solutions [18,63]. Furthermore, mapping the exposure of population involved according different level of temperature (LST) gain permits to address greening actions and policies, favor the identification of new medical or health centers, know in advance the areas that will be more subject to emergency calls, redevelop areas or zones most at risk with a view to mitigation and above all adaptation, make forecasts on access to hospitals in case of heat waves and the impact of costs on the health sector having mapped data, evaluate the effectiveness of requalification policies and actions and its ef-fects on heat flows and on the risk associated with the exposed population. Then, favor the development of new applications and services in a perspective technological transfer also to other sectors.

1.5 Aims

Finally, the main aim of this work was to perform a risk population assessment to rising temperatures and heat-waves by Landsat Land Surface Temperature timeseries in Aosta Valley, NW Italy by realizing a scalable application to each 18 countries that al-ready have HDX Meta dataset. The analysis was performed at a pixel level grouping the final population exposure at a municipality level. It is worth noting that the map generat-ed will be available at a pixel-level. Moreover, the quality of the population dataset was checked and a risk map performed considering the population distribution and the LST gains modelized. In particular, LST maximum and mean trends were computed consid-ering their significance and possible correlations were tested considering the following parameters: a) Fractional Vegetation Cover (FVC), b) Quote, c) Slope, d) Aspect e) Potential Incoming Solar Radiation (mean sun light duration in the meteorological summer season) in order to assess which parameters include into the risk model.

The final outputs have permitted to map and assess the LST gain in the last 39 years (1984-2022) and relative population exposure to LST trends per ages bands and gender groups providing hopefully useful information to civil protection and the health sector permitting them to detect areas in which call to health emergencies would be more likely during heatwaves and urban planners to promote greening actions in a mitigation and adaptation perspective to climate change according a One Health approach.”

  • As wisely suggested by the referee Fig.1 has been changed and improved taking into accountant also feedback from other reviewers so that it is now different from Figure 3. Please see into the manuscript.

  • We have moved table 5 to the appendix as suggested. The gaps between tables are related to the fact they represent different structural parameters of the population (gender, ages as reported in each caption section per each table). Please see into the manuscript.

  • We have kept them because they are simply images and just snapshot to see in more detail the exposure risks map moreover other referee suggest to keep them in the present form.

  • As suggested by the reviewer we have improved the discussion including the limitations. The validation concerning the risk cannot be performed because it is probabilistic. The ratio of exposure has been mapped as reported. Please tell us what do you mean for validation in case of probabilistic data? Moreover, it is worth noting that, LST is different from air temperature and cannot be compared. Please see into the manuscript we report some changes in the discussion here below: “Currently the only scientific mission with a higher resolution thermal sensor ECOS-TRESS on board the International Space Station does not allow long-term studies and was mainly designed as a tool for monitoring vegetation. Unfortunately, other scientific mis-sions that make satellite data available free of charge such as the European space program Copernicus have thermal data that is not suitable for conducting detailed studies. In fact, Sentinel-3 have a GSD of 1 Km. The development of high thermal resolution sensors with free access data would be desirable. Only the Albedo company is currently investing in high-resolution commercial satellite data as previously said, but it is not yet known whether its distribution policies will be free for research. However, in a mitigation and adaptation perspective to climate change, their implementation is not only desirable on a global level but also strategic in defining concrete One Health actions [77–79]. It is worth noting that, mapping the exposure of population involved according different level of temperature (LST) gain permits to address greening actions and policies, favor the identi-fication of new medical or health centers, know in advance the areas that will be more subject to emergency calls, redevelop areas or zones most at risk with a view to mitigation and above all adaptation, make forecasts on access to hospitals in case of heat waves and the impact of costs on the health sector having mapped data, evaluate the effectiveness of requalification policies and actions and its effects on heat flows and on the risk associated with the exposed population. Nevertheless, at the present time, analyzes of exposures to thermal trends are linked to EO data with GSD at 30 m (natively 100 m resampled at 30 m in case of Landsat). They currently represent the highest resolution available for scientific purposes. The hope is that the missions of the private company Albedo which will pro-vide satellite thermal data at 2 m GSD will be free for scientific purposes and will allow a significant technological and application leap. An increasingly detailed population da-taset is also desirable, although the aggregate Meta dataset is currently the most detailed from a spatial point of view. To date, in fact, although the present application is notable, it still limits the analyzes at a cluster level by areas, making analyzes at a building level more complex, which would certainly be desirable for the future. In fact, mapping the risk at the level of a single residential structure and its surroundings would allow increasingly precise actions with a view to adaptation and capillary and punctual analysis of the risk.

Concerning on the present study is interesting to see how the bottom of the Valley is more affected by rising temperature and how FVC do not play a statistically significant role (probably this is due to the fact that Landsat pixel GSD does not permit to appreciate in urban areas the effect offered by sparse vegetation (that normally considering the study area is less then, half pixel). It is interesting to know how the most of the highly risk areas are located in industrial areas and in modern buildings rather than in historical build-ings. At the same time, it is interesting to underline from a civil protection perspective how more than 60% of the Aosta Valley population (mostly concentrated in the valley floor for work reasons) is in high exposure and risk classes with both approaches adopted. Alt-hough in the summer some prefer to find refuge in the side valleys, the fact that a large part of the mostly elderly population is exposed to a greater risk should lead to rethinking urban planning and the creation of services or assistance hubs in areas with greater ex-posure. We hope a major application of EO Data in a One Health perspective [80].

Regardless of the considerations on the planning developments of climate adaptation and mitigation, we hope that this study will be useful and can also be scaled to other real-ities and become more and more detailed.”

Concerning Minor comments

  • L84: "the native geometric resolution for the thermal bands of the Landsat missions" has been rearranged in order to be clear. Please see into the manuscript.

  • While concerning, many abbreviations are used but never expanded when it was used for first time, the referee is right we have adjusted them into the text.

Author Response

Response to Reviewer 3 Comments

We would like to thank reviewers for their appropriate comments and helpful suggestions that have been carefully considered. Majority of provided suggestions highlighted gaps in the text and were really useful to improve, we hope, paper quality. Referees can find their comment and authors’ actions to reply/satisfy requests.

In particular, the synthesis of reviewers’ comments suggested a deep revision in paper organization and harmonization. Consequently, you will find some structural changes aimed at simplifying paper reading and make content more effective.

All comments were carefully evaluated and for the most of them corrections and integrations have been provided. Thank you so much for your work!

Point 1: Summary:

It seems the topic is interesting. However, I cannot complete reading the manuscript because at

certain point, I lost my confidence on the manuscript for publishing in any journal in current format.

With heavy modification, this work may be considered for publishing in a journal including Remote

Sensing.

It is hard for me to follow the description of the authors. I understand that the authors mainly used

GEE for the work, and the usage of GEE is different from the traditional way (data and algorithms

on a local platform). Nevertheless, I still like to see which parts of the algorithms are provided by the

authors to the GEE platform and the exact data (bands of Landsat) were processed with those

algorithms. Otherwise, my feeling is that the authors only need to configure the GEE to get the LST

time series without much inputs, and even in that case, what are the parameter settings. In short,

assuming I like to follow the work, the information provided by the authors should allow me to

repeat their work, at least in principle.

I think the manuscript is unnecessarily long and the logic connections are sometime loose (see major

comments below).

Major Comments:

  1. L51-4 (Lines 51 to 54): I believe the citations suggest just the opposite from what the authors

claimed.

  1. L108-128: The two paragraphs are not well organized. The overall tones are either repeating or

conflicting.

  1. L143-150: The sentence is simply too long. Consider dividing it into more sentences.
  2. Figure 1:
  3. The figure is not correctly cited in the main text.
  4. The display should be improved:
  5. It is not helpful to show the details of the area without much other information.

I even do not understand the meaning of the colors (or even the gray scales).

  1. I think the authors chose a very special map projection for showing the

locations of the special area in Italy. That is not very helpful to readers.

  1. I do not think the lists in L174-8 and L187-91 are very useful. Too much repeating information

in the lists. A better way is to condense the meaningful information and display the information

in a precise/concise table. In addition, I do not know which data sets provide the data needed to

calculate the NDVI and FVC.

  1. L206 (Equation 3):
  2. The authors should tell reader what epsilon means (emissivity) not matter how obvious

it is.

  1. The sentence structure for this equation does not follow the standard for a publication.
  2. I doubt the correctness of the equation although I do not have any knowledge on this

equation. The problem is with the parenthesis boundaries. Is ‘εs’ a subscript only or a

variable? If a variable, what it is? If not, the equation is not right.

2

  1. L132-4: The time gap is large from Nov. 2011 to March 2013. I like to know mor details on the

interpolation because the seasonal variations in the period.

  1. L250-89: This is an extremely long paragraph. However, I think most of the contents should be

removed from the manuscript. The authors used Meta data for their analysis only and they do

not need to tell the readers so much details on how to generate those data. Instead, they should

give more useful information about the data such as sources (links), data format/model,

resolution (mentioned), updating frequency, value ranges, any missing values, and uncertainty

(or accuracy, mentioned). The current description is distracting.

  1. Table 1: The caption and the headers sound strange to me. The problem is the word “structure.”

I do not think the table gives any information on data structure or physical structure. I guess

that authors meant “population structure” (population in age/gender groups).

  1. Section 2.3.1 and Equation (3) and corresponding discussions around that equation: I am

surprised that the authors suddenly mentioned that FVC is irrelevant with Landsat data in

Section 2.3.1. With Equation (3), I have been expecting that the variable values to be derived

from Landsat data.

  1. Table 2: I do not understand the role of this table. What is the usefulness of the information in

the table? And where?

  1. L347: What is the role for the Kolmogorov-Smirnov test?
  2. L349: what is the role of the Pettitt test?

I am sorry to say that at this point, I lost my faith on the quality of this manuscript and stopped here.

Minor Comments:

  • L17: I do not think the words in “Observation Data” should have any capitalized letter.

I think this problem appears in other places, too.

  • L19-20: This sentence is not logically connected to the first two sentences. Suggest modifying

the sentence.

  • L75: Please add a period before “However.”
  • L98: Please add a period before “However.”
  • In many places, the authors used “past years.” I feel strange when read this phrase. Suggest

changes.

  • L156: “Mars”?
  • L159-60: Please double check the punctuations for figure captions.
  • L162: The first part of the sentence sounds very strange.
  • L251: I finally see the authors give details on Meta. The details should be provided in the first

appearance of “Meta.”

  • L301 and 305: Table 3 mentioned before Table 1.
  • L310: Please add “to” before “develop.”
  • L315: The letter order is with an error.
  • L336: I do not like the phrase “here below it.” Actually, I do not like the phrase “per each” in

some places, either.

3

  • L340-1: I do not understand the logic in this sentence.

Response 1: Firstly, we would like to thank the reviewer for his/her comments.

Concerning on the first comment we are only partially agreed with the referee. We can try to understand the concerns raised but the algorithm has not been reported because it was developed by other authors and reporting it in its entirety would have been plagiarism. All the references of the case have been reported in the manuscript for further information and unlike what is reported, an expected reading allows you to follow all the input variables used and related settings in this regard, please refer to section 2.2. As reported in the text, the algorithm developed by Ermida also used by NASA itself allows starting from the brightness temperature bands of the various Landsat missions (band 6 in Landsat 4-5-7 and band 10 in Landsat 8-9) and by the NDVI to calculate LST for each available scenario. Moreover, this point represents a phase of the work that focuses on modeling trends and mapping exposed areas and it is risky and unfortunate that the review has been made without taking all aspects into account. For further info on the algorithm I will provide here a reference present into the manuscript: https://doi.org/10.3390/rs12091471

While concerning the fact that the manuscript is unecessarily long and the logic connections are sometime loose we agree with the reviewer. Therefore, we have adjuested the manuscript and move the tables in appendix and well organized the manuscript in order to be more clear hopefully. Please see the manuscript.

Major Comments

  1. We have just reported the population dataset… In all cases all the authors stated advances in new higher resolution population data would be nice. Although in their articles they suggest that they are high resolution datasets they are not, being currently the Meta one to have GSD at 30 m. It is clear that depending on the year of publication, development means that certain statements are no longer the same. And they were quoted correctly based on their products and datasets
  2. We have adjusted this part.
  3. We have adjusted this part and divided the introduction in sub section.
  4. As wisely suggested we have changed figure 1 and better contextualized the area in Italian territory. Moreover it has been cited into the text.
  5. We are partially agree with the reviewer we have changed a little bit and better explained which datasets have been used to compute NDVI and FVC. Please see into the manuscript.
  6. We have explained that episolon is the emissivity as you can see into the manuscript. The equation is correct and as you well understood is εs and εv are subscript of FVC as well reported below to avoid misunderstadings.
  7. An yearly interpolation was perfromed as repoted into the manuscript the gain was from yearly composites as reported into the manuscript “Landsat data were analyzing starting from 1984 till 2022 including therefore, 39 years of Landsat data. All acquisitions have been considered with more then, 900 images and bi-directional reflection disturbance compensated with a self-made function in GEE accord-ing to [69] regarding the NDVI. Clouds, shadows and saturated pixels have been masked out by considering Pixel Quality and Radiometric Saturation layers for each scene. Since the merged Landsat collections were not equally distributed through the time and there-fore not suitable to perform timeseries analysis due to temporal gaps all data have been filtered with a Savitzky-Golay filter [70–72] and regularized with a monthly timestep on GEE by adopting Open Earth Engine Library (OEEL). It is worth noting 2012 year was de-rived after creating yearly composite through linear interpolation as explained below.

Landsat images after November 2011 (last acquisition by Landsat 5) and before March 2013 (beginning of Landsat 8 mission) were retrieved by interpolation and regularization due to the lack of images during the period above mentioned. Landsat 7 data starting from 31 May 2003 onwards were not considered and therefore included in the timeseries regu-larization phase due to the failure of the Scan Line Corrector (SLC) which have affected the usage of these images. The correction of SLC was not performed in ENVI tool, despite there is an algorithm able to do because we processed them into GEE.

Then, yearly composite images were computed per each year in the time range 1984-2022 by using the ee.reducer GEE function in order to obtain the mean and the maximum pixel value per each year.”

  1. We agree with the reviewer therefore we have adjusted this part into the text anyway we have let a description on how Meta population dataset was obtained because it is impornat to know in absence of a scientific article how it was built as follow: “HDX Meta Facebook Population dataset was obtained as follow. Under the assump-tion that buildings act as a proxy for where people live, Meta (previously known as Face-book) obtains population estimates on a country-wide level, with 1x1 arcsecond resolution ( 30 × 30 m at the equator) and sensitivity to individual buildings, enabling accurate studies of population aggregation in rural areas. To enable global analysis Meta has de-veloped a building detection model. Meta pipeline consists of several steps: extraction of 64x64 pixel images (patches) around detected straight lines using a conventional edge de-tector, which reduces the amount of data for classification by 4 times. A portion of those candidates are sampled across all countries and labeled as training and evaluation data for the CNNs. The computer vision models are trained on a single machine with four GPUs, whereas the classification runs on Meta Facebook’s infrastructure on a CPU cluster. During this phase three different types of CNN were used: a classification model based on the SegNet [50]; a feedback neural network (FeedbackNet) performing weakly-supervised segmentation of the satellite images enabling Facebook to obtain building footprints [50], and a denoising network which is capable of improving the quality of the source data by removing high-frequency noise from the satellite imagery. The encoder-decoder style Se-gNet, is customized to perform the classification at the level of a patch. The encoder (a convolutional sub-network) is used to extract abstract information about the input, and the decoder (a deconvolutional sub-network) is trained to upsample the output of the en-coder into a spatially meaningful probability map representing the possibility of house existence in the input. The probabilities generated by the decoder are averaged over all spatial locations within the patch to derive the final classification including GNSS track-ing. This yields high accuracy and a reduced false positive rate on a global scale com-pared to other methods. To facilitate a generalized and scalable model, Meta employs the weakly-supervised learning that takes the abundant and easy-to-get image level categori-cal supervision (binary labeling) into training, and perform pixel-level prediction during deployment [50]. The methodology is motivated by the feedback mechanism in human cognition and recent advances of computational models in Feedback Neural Networks [50], which deactivates the non-relevant neurons within hidden layers of neural networks and achieve pixel-wise semantic segmentation. Both models are trained on 150,000 binary labeled (building/no building) patches, randomly sampled from all countries and sea-sons, covering both rural areas and urban areas. The output layer was validate consider-ing census at a country level reaching a global overall accuracy of 98.3% [50]. Then these data have been yearly coupled with aggregated tracking from Meta phone application (like Instagram, Facebook, WhatsApp). This data can be reachable through Meta Data for Good (https://dataforgood.facebook.com/ last access 19 April 2023), the format is raster (30 m GSD) or a dataframe and the updating frequency is yearly or more under request. In each dataset the pixel-value reports the population number according to a given charac-teristics.

To test the quality of Meta population 2020 product in a rural and mountain area like Aosta Valley Autonomous Region, in the North West of Italy, this dataset was tested con-sidering 2020 census at municipality level in Aosta Valley. HDX Meta Population dataset was considered as predicted population while regional census as observed true popula-tion. For each municipality in Aosta Valley the Mean Absolute Error (MAE) was computed”

  1. The referee is right we meant population structure and we have corrected.
  2. FVC was derived from Landsat Data for the thermal calibration if you read carefully. The FVC derived from Sentinel mentionend in section 2.3.1 was used only to assess vegetation cover precentage and obtain a layer as input in the risk modeler as better explain in this section and into the manuscript as follow: “Fractional Vegetation Cover was computed from Landsat Data to calibrate the LST as pre-viously reported, moreover to define the vegetation percentage in a single present layer used as possible input into the risk model, FVC was also estimated in ESA SNAP 8.0.0 open-source software starting from Copernicus Sentinel-2A (S2A) images. In particular a mean composite multi-band image in the 2020 summer meteorological season (from 1 June to 31 August) was generate in GEE with the function .mean() after applying cloud and shadows masking and the bidirectional reflectance distribution function (BRDF) with a self-made algorithm implemented in GEE. The composite output was exported from GEE preserving native resolution of each S2A band and then processed in ESA SNAP 8.0.0 by applying the Biophysical Processor S2_10m function. FVC was considered into the model in order to assess if vegetation may have a mitigation effect in temperature trends [8].” 
  3. It is the input parameters used to performed the Potential Incoming Solar Radiation analysi and obtain the map. This table is crucial to permit scalability!
  4. Before performing each kind of statistical analysis and modeling it is necessary to understand if data follow a normal distribution or not. This is basic best practise statistics and we have reported a test of normality in this case Kolmogorov. We have preferred reported into the manuscript to be rigorous.
  5. Pettitt’s test was carried out because, as indicated, only the significant pixels were modeled, therefore, thanks to it, break points were identified in the time series.

Hope you will have faith now on the work performed.

Minor Comments

  • Capitalization is used because it refers to Earth Observation Data generally indicated as capitalized in all papers!
  • The reviewer is right we have adjusted by deleting therfore.
  • The reviewer is right we have corrected please see into the manuscript.
  • The reviewer is right we have corrected please see into the manuscript.
  • We have changed past with last.
  • It is an error we have corrected with March
  • L159-60: Please double check the punctuations for figure captions. Done
  • ­ L162: The first part of the sentence sounds very strange. We have corrected.
  • Adjusted the Meta dataset description please see into the manuscript.
  • Yes it is correct we have added see Appendix A.
  • Please add “to” before “develop.” Done
  • Thanks corrected.
  • We have changed the phrase here below into the manuscript as well as per each pelase see into the manuscript.
  • We have rewritten the sentence as follow: “To detect surface height the regional deep learning dataset realized in 2020 was adopted”

Reviewer 4 Report

The manuscript presents interesting research findings; however, I suggest that the authors revise and enhance two aspects of their work before considering it for publication. Firstly, the introduction requires improvement in terms of emphasizing the research purpose and significance. Presently, the authors have not provided a comprehensive description of the significance of their work. Secondly, the discussion section needs to address the limitations and prospects of the study, which is crucial for a thorough analysis of the research outcomes. Additionally, the research methods presented in the manuscript are too convoluted and require refinement to enhance clarity and readability. By addressing these issues, the manuscript will provide a valuable contribution to the field and meet the standard for publication.

The overall English expression is okay, and there are no major grammatical errors.

Author Response

Response to Reviewer 4 Comments

We would like to thank reviewers for their appropriate comments and helpful suggestions that have been carefully considered. Majority of provided suggestions highlighted gaps in the text and were really useful to improve, we hope, paper quality. Referees can find their comment and authors’ actions to reply/satisfy requests.

In particular, the synthesis of reviewers’ comments suggested a deep revision in paper organization and harmonization. Consequently, you will find some structural changes aimed at simplifying paper reading and make content more effective.

All comments were carefully evaluated and for the most of them corrections and integrations have been provided. Thank you so much for your work!

Point 1: The manuscript presents interesting research findings; however, I suggest that the authors revise and enhance two aspects of their work before considering it for publication. Firstly, the introduction requires improvement in terms of emphasizing the research purpose and significance. Presently, the authors have not provided a comprehensive description of the significance of their work. Secondly, the discussion section needs to address the limitations and prospects of the study, which is crucial for a thorough analysis of the research outcomes. Additionally, the research methods presented in the manuscript are too convoluted and require refinement to enhance clarity and readability. By addressing these issues, the manuscript will provide a valuable contribution to the field and meet the standard for publication.

Response 1: Firstly, we would like to thank the reviewer for his/her valuable comments and suggestion. As wisely suggested we have improved the manuscript in particular: we have improved the introduction emphasizing the research purpose and significance. To do so we have included new parts and divided the introduction in subsection to help the readers. Here we report the introduction: “Temperatures and Summer heatwaves monitoring due to ongoing climate change has assumed a crucial role in the last years worldwide [1–3]. Although studies on extreme events are increasing and in particular on heatwaves and Urban Heat Island [4–8], few focuses on time series derived from free Earth Observation images [4,9–15]. Furthermore, there is still a lack regarding to scientific and technical studies that focus on Land Surface Temperature (hereinafter called LST) climatic trends through an analysis of the exposed population [16–18].

Nowadays, many studies focused on LST and epidemiological relationships but do not concern spatial population exposure [19–21] or animals including wildlife [22].

1.1 Earth Observation (EO) Data role in the climate change framework

The evaluation of exposure to ambient temperatures in epidemiological studies has generally been based on records from meteorological stations which may not adequately represent local temperature variability [23].

To go beyond this limiting factor, Earth Observation images represent a possible so-lution to carefully map environmental condition at local scale [20,24]. The health sector and civil protection services in recent years at European, Italian and local level are partic-ularly interested in having cartographic products and GIS to assess the risks and effects of extreme temperatures on the population by identifying the most vulnerable areas [19]. The identification of these areas would make it possible to direct territorial planning towards greening policies or measures aimed at mitigating warming and at the same time imple-menting forms of adaptation (for example, creation of emergency response hubs in the case of an area with a vulnerable population such as the elderly). Although free thermal data is increasing by offering medium-high spatial resolution (like Landsat missions [25,26] with a resample GSD 30 m or ECOSTRESS with 60 m GSD [27–32], their use for the development of various services and applications is still limited [33–36] and therefore, of-fers numerous exploitation possibilities when combined with new databases made available by various governmental or research bodies.

Thermal data are widely applied nowadays to map LST and urban heat island phe-nomenon [8,37–41]. However, their use is often confined to analysis in given moments and not in timeseries due to the need to calibrate them [34,42]. Platforms such as Google Earth Engine [43] in the case of Landsat data allows thanks to the algorithm developed by Ermida [36] to quickly calibrate the thermal data allowing analysis on historical series.

1.2 Population datasets

In recent years, datasets on the spatial and temporal distribution of the global popu-lation have been developed [44–46]. Nevertheless, they still have a coarse resolution. One of the most detailed is provided by the World Bank with the World Population dataset with a spatial resolution of 1Km and another to 100 m. This last is spatially coeval with the native geometric resolution for the thermal bands of the Landsat missions [46]. This dataset contains top-down constrained breakdown of estimated population by age and gender groups from 2000 to present year [45]. Top-down constrained age/sex structure es-timate datasets for individual countries for 2020 at 100m spatial resolution with country totals adjusted to match the corresponding official United Nations population estimates that have been prepared by the Population Division of the Department of Economic and Social Affairs of the United Nations Secretariat (2019 Revision of World Population Pro-spects). It is worth to note that WorldPop gridded datasets on population age structures, poverty, urban growth, and population dynamics are freely available. Despite of the huge amount of data this dataset still have limited application in rural context and outside wide urban areas due to its geometric resolution that limited the application at regional and local level [47–49] .

Accurate information on global population distribution is crucial to many disci-plines. A population and housing census are the traditional tool for deriving small-area detailed statistics on population and its spatial distribution [50,51] However, censuses are time-consuming, and the spatial resolution is naturally set by the census enumeration ar-eas (EA), which lack fine-grained information about the aggregation of population. The sizes of the EAs vary by many orders of magnitude from country to country, ranging from hundreds of square meters in urban areas to tens of thousands of square kilometers in low population areas, resulting in an average spatial resolution [50] of a census unit of 33 km at a global scale. Recently, multiple higher resolution maps of human-made built up areas have emerged [52,53], most notably the Global Human Settlement Layer (GHSL) [54], the Global Urban Footprint (GUF) [51,55] , the WorldPop project [44,56], Landscan [57,58] and Missing Maps project [59,60]. However, none provide a scalable solution with high accu-racy in rural areas. Over the past decade high-resolution (sub-meter) satellite imagery has become widely available, enabling the global collection of recent and cloud-free earth im-agery. Additionally, in the past years, the surge in research on computer vision and in particular convolutional neural networks (CNN) have enabled bulk processing of imagery in a rapid manner [50]. The combination of these methods enables the global analysis of high-resolution imagery as a promising method for detecting individual buildings; com-bining building estimates with available census data to produce updated and higher res-olution population maps; and offering alternative, state-of-the-art population estimates in the absence of census data. Various approaches using machine learning have been demonstrated on small areas [50], yet a method which allows global mapping has re-mained elusive.

Nowadays, in fact, high resolution datasets of population density which accurately map sparsely-distributed human populations do not exist at a global scale [49,50,61]. Typically, population data is obtained using censuses and statistical modeling. More re-cently, methods using remotely-sensed data have emerged, capable of effectively identify-ing urbanized areas. Obtaining high accuracy in estimation of population distribution in rural areas remains a very challenging task due to the simultaneous requirements of suffi-cient sensitivity and resolution to detect very sparse populations through remote sensing as well as reliable performance at a global scale. Meta has recently developed a computer vision method based on machine learning to create population maps from satellite im-agery and phone GNSS tracking at a global scale, with a spatial sensitivity corresponding to individual buildings and suitable for global deployment. By combining this settlement data with census data, Meta has created HDX Meta population dataset including raster maps with ~30 meter spatial resolution for 18 countries in the World [50]. HDX is a plat-form which lets users for research and management purposes access socio-economical data mostly collected by Meta through Data for Good (https://dataforgood.facebook.com/dfg last access 18/04/2023). Data for Good at Meta's program includes tools built from de-identified Meta data, as well as tools that the com-pany develops using satellite imagery and other publicly available sources.

1.3 Remote sensing in climate change risk assessment

There is a growing need for the assessment and reduction of climate change risk. The effects of global warming are already bringing harm to human communities and the nat-ural world. Further temperature rises will have a devastating impact and more action on greenhouse gas emissions is urgently required. Multiple factors contribute to climate change, and multiple actions are needed to address it [8,37–41]. Especially concerning on population exposure to climate change. In fact, nowadays, EO Data and more generally remote sensing may help in mapping and developing services with particular regard onto climate change risk assessment involving communities at different level. Space-borne images for civil applications have been routinely acquired since the 1980s (Landsat and SPOT), while more recently, the European Union’s Copernicus program. EO data can pro-vide remotely sensed information regarding floods, forest fires, and droughts. In general, remote sensing data from space, but also from airborne or drone platforms, can be profita-bly used to manage many risks, from geo-hydrological to volcanic, from seismic to an-thropogenic. Less exploited is the application and coupling of remote sensing data with GIS health data with particular regard onto the climate change framework. Remote sens-ing can play a key role in managing risks, leading to a new level of understanding of the complex Earth systems and planning. In recent decades, satellite-based observations and the derived geospatial products have been successfully demonstrated to be highly valua-ble tools in each different phase of the risk and exposure management (forecasting, plan-ning, emergency, and post-emergency) [34,42]. For example, synthetic aperture radar (SAR) can facilitate risk management since they are also acquired through dense cloud cover and in both night and day conditions. This ability can help during the emergency phase. Stacks of SAR data can be used to detect subtle ground deformation induced by slow movement phenomena (e.g., slow landslides, subsidence) that can dangerously evolve, involving elements of risk [62]. On the other hand, optical images are fundamental products to monitor land cover changes induced by several hazards (e.g., fast landslides, volcanic eruptions) or thermal data to assess for eg. Urban heath island (UHI) and their intensity or the water stress onto forests and crops. These data are routinely used to map and evaluate the element at risk scattered over wide areas. The application of a combined used of population data at higher resolution with thermal EO Data in order to evaluate the exposure to rising temperature in light of climate change has poorly explored in the scien-tific community. This is due to the fact that the population dataset at higher resolution are relatively new and also the application of EO Data in the domain. of the climate change adaptation regarding the civil component are moving their first steps.

Moreover, the One Health approach involving thermal remote sensed time-series analysis to assess the temperature trend gain represent a novelty than the well-know and over-studied UHI phenomenon which is focused only on a given time and do not permit to develop strong models. The LST trends analysis modelling and their application cou-pling population data represent a novelty especially in the assessment of rising tempera-ture exposure [8].  

1.4 Coupling population and EO Data in climate change adaptation and risk assessment

The approach developed to map population (thanks to Meta Geo For Good) coupling free thermal EO Data to model rising temperature in order to map the exposure and risk considering population age bands and gender groups with all dataset involved with the same native geometrical resolution (GSD 30 m) enforces the applicability and coupling of these kind of data in the planning and management of climate change risks and adapta-tion suggesting new solutions [18,63]. Furthermore, mapping the exposure of population involved according different level of temperature (LST) gain permits to address greening actions and policies, favor the identification of new medical or health centers, know in advance the areas that will be more subject to emergency calls, redevelop areas or zones most at risk with a view to mitigation and above all adaptation, make forecasts on access to hospitals in case of heat waves and the impact of costs on the health sector having mapped data, evaluate the effectiveness of requalification policies and actions and its ef-fects on heat flows and on the risk associated with the exposed population. Then, favor the development of new applications and services in a perspective technological transfer also to other sectors.

1.5 Aims

Finally, the main aim of this work was to perform a risk population assessment to rising temperatures and heat-waves by Landsat LST timeseries in Aosta Valley, NW Italy by realizing a scalable application to each 18 countries that already have HDX Meta da-taset. The analysis was performed at a pixel level grouping the final population exposure at a municipality level. It is worth noting that the map generated will be available at a pix-el-level. Moreover, the quality of the population dataset was checked and a risk map per-formed considering the population distribution and the LST gains modelized. In particu-lar, LST maximum and mean trends were computed considering their significance and possible correlations were tested considering the following parameters: a) Fractional Veg-etation Cover (FVC), b) Quote, c) Slope, d) Aspect e) Potential Incoming Solar Radiation (mean sun light duration in the meteorological summer season) in order to assess which parameters include into the risk model.

The final outputs have permitted to map and assess the LST gain in the last 39 years (1984-2022) and relative population exposure to LST trends per ages bands and gender groups providing hopefully useful information to civil protection and the health sector permitting them to detect areas in which call to health emergencies would be more likely during heatwaves and urban planners to promote greening actions in a mitigation and adaptation perspective to climate change according a One Health approach.”

Concerning, the discussion section it has been included the limitations and prospects of the study, as it follows: “…Currently the only scientific mission with a higher resolution thermal sensor ECOS-TRESS on board the International Space Station does not allow long-term studies and was mainly designed as a tool for monitoring vegetation. Unfortunately, other scientific mis-sions that make satellite data available free of charge such as the European space program Copernicus have thermal data that is not suitable for conducting detailed studies. In fact, Sentinel-3 have a GSD of 1 Km. The development of high thermal resolution sensors with free access data would be desirable. Only the Albedo company is currently investing in high-resolution commercial satellite data as previously said, but it is not yet known whether its distribution policies will be free for research. However, in a mitigation and adaptation perspective to climate change, their implementation is not only desirable on a global level but also strategic in defining concrete One Health actions [77–79]. It is worth noting that, mapping the exposure of population involved according different level of temperature (LST) gain permits to address greening actions and policies, favor the identi-fication of new medical or health centers, know in advance the areas that will be more subject to emergency calls, redevelop areas or zones most at risk with a view to mitigation and above all adaptation, make forecasts on access to hospitals in case of heat waves and the impact of costs on the health sector having mapped data, evaluate the effectiveness of requalification policies and actions and its effects on heat flows and on the risk associated with the exposed population. Nevertheless, at the present time, analyzes of exposures to thermal trends are linked to EO data with GSD at 30 m (natively 100 m resampled at 30 m in case of Landsat). They currently represent the highest resolution available for scientific purposes. The hope is that the missions of the private company Albedo which will pro-vide satellite thermal data at 2 m GSD will be free for scientific purposes and will allow a significant technological and application leap. An increasingly detailed population da-taset is also desirable, although the aggregate Meta dataset is currently the most detailed from a spatial point of view. To date, in fact, although the present application is notable, it still limits the analyzes at a cluster level by areas, making analyzes at a building level more complex, which would certainly be desirable for the future. In fact, mapping the risk at the level of a single residential structure and its surroundings would allow increasingly precise actions with a view to adaptation and capillary and punctual analysis of the risk.

Concerning on the present study is interesting to see how the bottom of the Valley is more affected by rising temperature and how FVC do not play a statistically significant role (probably this is due to the fact that Landsat pixel GSD does not permit to appreciate in urban areas the effect offered by sparse vegetation (that normally considering the study area is less then, half pixel). It is interesting to know how the most of the highly risk areas are located in industrial areas and in modern buildings rather than in historical build-ings. At the same time, it is interesting to underline from a civil protection perspective how more than 60% of the Aosta Valley population (mostly concentrated in the valley floor for work reasons) is in high exposure and risk classes with both approaches adopted. Alt-hough in the summer some prefer to find refuge in the side valleys, the fact that a large part of the mostly elderly population is exposed to a greater risk should lead to rethinking urban planning and the creation of services or assistance hubs in areas with greater ex-posure. We hope a major application of EO Data in a One Health perspective [80].

Regardless of the considerations on the planning developments of climate adaptation and mitigation, we hope that this study will be useful and can also be scaled to other real-ities and become more and more detailed.”

Concerning the methods they have been adjusted in order to be much readable please see into the manuscript.

Reviewer 5 Report

Dear Authors,

I do really appreciate your work, in particular the innovative multi data source approach and their novel application. Great job.

In the paper are presents minor typos and very few things to correct, you will find easily. 

But some comments are necessary to improve your paper.

1) The LST is normally and more accurately estimated by using nighttime acquired images to avoid the effect of direct solar irradiation.  GEE allow to use only daytime data (Ermida 2020). Can I ask you to clarify in the text why you did not used Landsat nighttime collection. The use of boths dataset (day and night is also used to estimate the UHI effect)

2) Sentinel 2 and Landsat are only recently acquired in parallel, it wasn't true at the beginning of your time serie. therefore the FVC, i suppose, was estimated differently.

3) Landsat and Sentinel have some problem on coregistration, how did you fix it?

4)Despite the problem of increasing LST is evident you never presents relative or absolute value variation in the period you studied.  I think it could be useful for the sizing of the problem. And as a consequence how you validated the LST obtained.   

I would like to read your implemented response before publishing your paper.

Thanks

Author Response

Response to Reviewer 5 Comments

We would like to thank reviewers for their appropriate comments and helpful suggestions that have been carefully considered. Majority of provided suggestions highlighted gaps in the text and were really useful to improve, we hope, paper quality. Referees can find their comment and authors’ actions to reply/satisfy requests.

In particular, the synthesis of reviewers’ comments suggested a deep revision in paper organization and harmonization. Consequently, you will find some structural changes aimed at simplifying paper reading and make content more effective.

All comments were carefully evaluated and for the most of them corrections and integrations have been provided. Thank you so much for your work!

Point 1: I do really appreciate your work, in particular the innovative multi data source approach and their novel application. Great job.

In the paper are presents minor typos and very few things to correct, you will find easily.

But some comments are necessary to improve your paper.

1) The LST is normally and more accurately estimated by using nighttime acquired images to avoid the effect of direct solar irradiation.  GEE allow to use only daytime data (Ermida 2020). Can I ask you to clarify in the text why you did not used Landsat nighttime collection. The use of boths dataset (day and night is also used to estimate the UHI effect)

2) Sentinel 2 and Landsat are only recently acquired in parallel, it wasn't true at the beginning of your time serie. Therefore, the FVC, i suppose, was estimated differently.

3) Landsat and Sentinel have some problem on coregistration, how did you fix it?

4)Despite the problem of increasing LST is evident you never presents relative or absolute value variation in the period you studied.  I think it could be useful for the sizing of the problem. And as a consequence how you validated the LST obtained.  

I would like to read your implemented response before publishing your paper.

Thanks

Response 1: Firstly, we would like to thanks the reviewer for his/her suggestion and comments. A general comprehensive editing has been done taking into account also the suggestion proposed by the other reviewers.  Concerning the specific comments:

  • We adopted the daytime data because we were interested in mapping the daily trends which represent the major risk. We are aware of this but we focus our attention onto the maximum LST as explained into the manuscript. We have gone beyond the UHI effects mapping and measuring a trend over 39 years and modelized the gain. We have explained as followed into the manuscript “It is worth noting that, LST is normally and more accurately estimated by using nighttime acquired images to avoid the effect of direct solar irradiation in case of study of UHI, nev-ertheless the present study has focused on LST maximum trends that normally occur during the day. Moreover, the risk and exposure to the population are higher during sun light. For these reasons this research has been focus on daytime LST.”

  • To calibrate LST FVC was computed using Landsat Data as reported into the manuscript. We used Sentinel-2 only to compute FVC at the present day to define the green areas used into the risk model. I hope you have now well understood.

  • I understood your question but see answer 2. FVC is a stand-alone product not involved in thermal processing in which FVC was computed using Landsat as better reported into the manuscript in order to avoid misunderstandings. We have also better explain in section 2.3.1 please see into the manuscript.

  • Concerning the orbit the LST collection were made by composite images therefore the concept of orbit no longer takes on a meaning because we are dealing with images with pixels derived from a multitude of observations within the same area. In this regard, for further clarification and explanation of the process, please refer to the link: https://developers.google.com/earth-engine/guides/ic_composite_mosaic

Concerning the validation the algorithm adopted and the workflow has been tested for the whole collection as reported in the manuscriopt and paper probably you lost them during the reading. To support this I report here the manuscript to validate the algorithm adopted and the relative collection in GEE:

- https://doi.org/10.3390/rs12091471

- https://doi.org/10.1016/j.rse.2014.03.016

-  https://doi.org/10.3390/rs10071114

Round 2

Reviewer 2 Report

After reading the revised manuscript, I find that the authors have addressed all comments and queries. The manuscript has improved significantly. Moreover, Fig 6 is not revised as I suggested to keep the lat and lon frame. It will be better it can be revised. 

NA

Author Response

Response to Reviewer 2 Comments

We would like to thank reviewers for their appropriate comments and helpful suggestions that have been carefully considered. Majority of provided suggestions highlighted gaps in the text and were really useful to improve, we hope, paper quality. Referees can find their comment and authors’ actions to reply/satisfy requests.

In particular, the synthesis of reviewers’ comments suggested a deep revision in paper organization and harmonization. Consequently, you will find some structural changes aimed at simplifying paper reading and make content more effective.

All comments were carefully evaluated and for the most of them corrections and integrations have been provided. Thank you so much for your work!

Point 1:  After reading the revised manuscript, I find that the authors have addressed all comments and queries. The manuscript has improved significantly. Moreover, Fig 6 is not revised as I suggested to keep the lat and lon frame. It will be better it can be revised.

Response 1: Firstly, we would like to thank the reviewer for his/her valuable comments. We have followed the last suggestion given and added in Fig 6 lat and lon frame. Please see into the manuscript.

Reviewer 3 Report

Summary:

I did not try to read the major parts of the manuscript again. I read the Abstract and the Introduction, and try to check the responses to my comments.

The Introduction is still not coherent leading to the study of this work. By a rough comparison between this version and the last version, I think the main changes are the adding of subsections, and the contents of Subsections 1.3 and 1.4. Be honest, it is very hard for me to read the Introduction with a logical idea flow. To me, it seems that the authors try to put a lot of not-very-relevant stuff together. In addition, there are still a lot of grammar issues with the Introduction (see Minor Comments).

The one-to-one responses are not all satisfied. For example, the authors answered my Points 12 and 13 in the Major Comments section. However, there is not reflection on the main text. The purpose for my comments is to make the manuscript more understandable. In current format, I do not think one can fully understand what the authors intend to do and how to do. Therefore, I still suggest a major revision.

Minor Comments:

·         L30-1: I do not think the sentence is a grammatically correct sentence.

·         L35-8: I do not think the sentence is a grammatically correct sentence.

·         L40-1: Please pay attention on the punctuations, not consistent and confusing.

·         As I mentioned in last round, the authors use the capitalized words arbitrarily. That is a minor issue but annoying for a formal publication. One example is L45, “Summer” and another example is “Urban heath island (UHI)” (L162) for which I think all three words should be started with a capitalized letter.

·         The authors still use a lot of acronyms without definition. I do not think it is necessary to define the widely-used ones such as GIS or NASA. However, for those not widely used, their appearance without definition will make the reading difficult. For example, I can guess the meaning of GSD (L68) for spatial resolution but I failed to guess the implied words by this acronym. I totally lost for the meaning of “CM” (L420).

·         L72-3: I am afraid this is a run-on sentence.

·         L95-8: I am afraid this is a run-on sentence.

·         L183-90: The sentence is too long, and I believe there is a grammatical error.

See the comment above. Improvement must be made.

Author Response

Response to Reviewer 3 Comments

We would like to thank reviewers for their appropriate comments and helpful suggestions that have been carefully considered. Majority of provided suggestions highlighted gaps in the text and were really useful to improve, we hope, paper quality. Referees can find their comment and authors’ actions to reply/satisfy requests.

In particular, the synthesis of reviewers’ comments suggested a deep revision in paper organization and harmonization. Consequently, you will find some structural changes aimed at simplifying paper reading and make content more effective.

All comments were carefully evaluated and for the most of them corrections and integrations have been provided. Thank you so much for your work!

Point 1: Summary:

I did not try to read the major parts of the manuscript again. I read the Abstract and the Introduction, and try to check the responses to my comments.

The Introduction is still not coherent leading to the study of this work. By a rough comparison between this version and the last version, I think the main changes are the adding of subsections, and the contents of Subsections 1.3 and 1.4. Be honest, it is very hard for me to read the Introduction with a logical idea flow. To me, it seems that the authors try to put a lot of not-very-relevant stuff together. In addition, there are still a lot of grammar issues with the Introduction (see Minor Comments).

The one-to-one responses are not all satisfied. For example, the authors answered my Points 12 and 13 in the Major Comments section. However, there is not reflection on the main text. The purpose for my comments is to make the manuscript more understandable. In current format, I do not think one can fully understand what the authors intend to do and how to do. Therefore, I still suggest a major revision.

Minor Comments:

  • L30-1: I do not think the sentence is a grammatically correct sentence.

  • L35-8: I do not think the sentence is a grammatically correct sentence.

  • L40-1: Please pay attention on the punctuations, not consistent and confusing.

  • As I mentioned in last round, the authors use the capitalized words arbitrarily. That is a minor issue but annoying for a formal publication. One example is L45, “Summer” and another example is “Urban heath island (UHI)” (L162) for which I think all three words should be started with a capitalized letter.

  • The authors still use a lot of acronyms without definition. I do not think it is necessary to define the widely-used ones such as GIS or NASA. However, for those not widely used, their appearance without definition will make the reading difficult. For example, I can guess the meaning of GSD (L68) for spatial resolution but I failed to guess the implied words by this acronym. I totally lost for the meaning of “CM” (L420).

  • L72-3: I am afraid this is a run-on sentence.

  • L95-8: I am afraid this is a run-on sentence.

  • L183-90: The sentence is too long, and I believe there is a grammatical error.

Response 1: Firstly, we would like to thank the reviewer for his/her comments despite it is not fair to judge a manuscript without a carefully read of the entire manuscript! It sounds strange that the reviewer is the only of 5 reviewers that does not find the manuscript logic and coherent. All the others as it is possible to see have positively judge the quality of the work and its logical structure. We have therefore followed the suggestion given partially and we apologize for this. Moreover, the complains are not supporting by useful suggestion as a peer-reviewers process needs to. We found more useful by providing motivated and objective suggestions and not simply generic comments which denote a certain mockery of the work which is otherwise not supported by scientific evidence and constructive criticism of the method. It is not fair according to us judging a paper without a deep read as reported into reviewer guideline of MDPI as well as any other Scientific Journal.

We are very sorry for this treatment. The introduction if you read the whole work is logical and follows a careful description of the whole workflow and it is clear that reading only pieces and judging it without a complete reading leads to making false judgments.

In any case we have replied to minor comments and detailed in the text the criticisms made in the previous revision explaining points 12 and 13 also in the manuscript. We thank for this points and suggestion.

Minor Comments

  • The reviewer is right the sentence has been corrected.
  • We have corrected.
  • Thank you done
  • Totally agree we have followed your suggestion concerning for. eg UHI
  • We agree with the reviewer GSD has been defined into the text while CM is already defined into the text as follow in section 2.4: “…In LST max and mean gain layers, pixels not significant were masked out considering only significant Pettitt’s p-value < 0.05 (here in after called CM) after clipping the data onto the Population dataset. Subsequently, ISODATA unsupervised classification - clustering algorithm was performed on CM and the separability of each class obtained was checked by computing the Jeffries-Matusita (J-M) distances…”
  • It is the first time that it appears in the introduction… Please be more specific
  • The same as above…
  • The referee is right we have changed as follow: “The approach developed to map population (thanks to Meta Geo For Good) coupling free thermal EO Data to model rising temperature represents a first attempt of this kind. In order to map the population exposure and risk for the first time the highest available pop-ulation dataset has been used with a native geometrical resolution (GSD – Ground Sample Distance 30 m) which is the same of Landsat. This may enforces the applicability and coupling of these kind of data in the planning and management of climate change risks and adaptation suggesting new solutions [18,63].”